



# Intercomparison and characterization of 23 Aethalometers under laboratory and ambient air conditions: Procedures and unit-to-unit variabilities

Andrea Cuesta-Mosquera[1], Griša Močnik[2,3,4], Luka Drinovec[2,3], Thomas Müller[1], Sascha Pfeifer[1], María Cruz Minguillón[5], Briel Björn[6], Paul Buckley[7], Vadimas Dudoitis[8], Javier Fernández-García[9], María Fernández-Amado[10], Joel Ferreira De Brito[11], Harald Flentje[6], Eimear Heffernan[7], Nikolaos Kalivitis[12], Athina-Cerise Kalogridis[13], Hannes Keernik[14,15], Luminita Marmureanu[16], Krista Luoma[17], Angela Marinoni[18], Michael Pikridas[19], Gerhard Schauer[20], Norbert Serfozo[21], Henri Servomaa[22], Gloria Titos[23], Jesús Yus-Díez[5,24], Natalia Zioła[25], Alfred Wiedensohler[1]

[1]Department of Experimental Aerosol and Cloud Microphysics, Leibniz Institute for Tropospheric Research, Leipzig, 04318, Germany
[2]Department of Condensed Matter Physics, Jožef Stefan Institute, Ljubljana, 1000, Slovenia
[3]Haze Instruments d.o.o., Ljubljana, 1000, Slovenia
[4]Center for Atmospheric Research, University of Nova Gorica, Ajdovščina, 5270, Slovenia
[5]Institute of Environmental Assessment and Water Research (IDAEA), CSIC, Barcelona, 08034, Spain
[6]Deutscher Wetterdienst (DWD), Meteorologisches Observatorium Hohenpeißenberg, Hohenpeißenberg, 82383, Germany
[7]School of Chemistry, Centre for Research into Atmospheric Chemistry & Environmental Research Institute, University College Cork, Cork, T23 XE10, Ireland
[8]Department of Environmental Research, SRI Center for Physical Sciences and Technology, Vilnius, 10257, Lithuania
[9]Centro de Investigaciones Energéticas, Medioambientales y Tecnológicas, Madrid, 28040, Spain
[10]Instituto Universitario de Medio Ambiente-Grupo QANAP, Universidade da Coruña, Oleiros, 15179, Spain
[11]IMT Lille Douai, Université de Lille, Lille, 59508, France
[12]Department of Chemistry, University of Crete, Heraklion, 70013, Greece
[13]Environmental Radioactivity Laboratory, National Centre of Scientific Research, Aghia Paraskevi, 15310, Greece
[14]Air Quality Management Department, Estonian Environmental Research Centre, Tallinn, 10617, Estonia
[15]Department of Software Sciences, Tallinn University of Technology, Tallinn, 12616, Estonia
[16]National Institute of Research and Development for Optoelectronics, Măgurele, 077125, Romania
[17]Institution for Atmospheric and Earth System Research, University of Helsinki, Helsinki, 00014, Finland
[18]Institute of Atmospheric Sciences and Climate, National Research Council of Italy, Bologna, 40129, Italy
[19]Climate, Atmosphere Research Centre (CARE-C), The Cyprus Institute, Nicosia, 1645, Cyprus
[20]Sonnblick Observatory, Central Institute for Meteorology and Geodynamics (ZAMG), Salzburg, 5020, Austria
[21]Global Change Research Institute, Brno, 60300, Czech Republic
[22]Finnish Meteorological Institute, Helsinki, FI-00101, Finland
[23]Andalusian Institute for Earth System Research, University of Granada, Granada, 18006, Spain
[24]Departament of Applied Physics - Meteorology, University of Barcelona, Barcelona, 08028, Spain
[25]Department of Air Protection, Institute of Environmental Engineering of the Polish Academy of Sciences, Zabrze, 41-819, Poland

*Correspondence to*: Andrea Cuesta-Mosquera (cuesta@tropos.de)

**Key words:** filter-based absorption photometer, aethalometer, intercomparison



**Abstract.** Airborne black carbon particles are monitored in many networks to quantify its impact on air quality and climate. Given its importance, measurements of black carbon mass concentrations must be conducted with instruments operating in a quality checked and assured conditions to generate reliable and comparable data. According to WMO (World Meteorological Organization) and GAW (Global Atmosphere Watch), intercomparisons against a reference instrument are a crucial part of quality controls in measurement activities (WMO, 2016).

The WMO-GAW World Calibration Centre for Aerosol Physics (WCCAP) carried out several instrumental comparison and calibration workshops of absorption photometers in the frame of ACTRIS (European Research Infrastructure for the observation of Aerosol, Clouds and Trace Gases) and the COST Action COLOSSAL (Chemical On-Line cOmpoSition and Source Apportionment of fine aerosoL) in January and June 2019.

The experiments were conducted to intercompare filter-based particle light absorption photometers, specifically aethalometers AE33 (Magee Scientific), which are operated by research institutions, universities or governmental entities across Europe. The objective was to investigate the individual performance of 23 instruments and their comparability, using synthetic aerosols in a controlled environment and ambient air from the Leipzig urban background. The methodology and results of the intercomparison are presented in this work.

The observed instrument-to-instrument variabilities showed differences that were evaluated, before maintenance activities (average deviation from total least square regression: 1.1%, range: -6% to 16%, for soot measurements; average deviation: 0.3%, range: -14% to 19%, for nigrosin measurements), and after they were carried out (average deviation: 0.4%, range: -8% to 14%, for soot measurements; average deviation: 1.1%, range: -15% to 11%, for nigrosin measurements). The deviations are in most of the cases explained by the filter material, the total particles load on the filter, the performance of the flow systems and previous flow check and calibrations carried out with non-calibrated devices.

The results of this intensive intercomparison activity show that relatively small unit-to-unit uncertainties of AE33-based particle light absorbing measurements are possible with functioning instruments. It is crucial to follow the guidelines for maintenance activities and the use of the proper filter tape in the AE33 to assure high quality and comparable BC measurements among international observational networks.

## 1. Introduction

The impact of black carbon (BC) on climate, health and human activities prioritizes the observation of BC mass concentration and its optical properties in different environments. In the atmosphere, BC absorbs solar radiation from the visible to the infrared optical spectrum, causing visibility degradation and making it the second most important radiative forcer (Ramanathan and Carmichael, 2008). Black carbon particles modify the lifetime, distribution, and formation processes of clouds, because they can act as cloud condensation nuclei, ice nuclei (predominantly in the cirrus temperature range), and modify clouds internal mixing state, therefore alters clouds albedo (Koch et al., 2011; Bond et al., 2013; Chen et al., 2018; Wex et al., 2019). BC is also well-known as an air pollutant, affecting human health since it serves as a carrier of multiple



toxic substances, which are harmful for the respiratory system, the cardiac function and the immune system (Janssen et al., 2011; WHO, 2012). In consequence, networks for observation of atmospheric black carbon are growing and need to be maintained worldwide. BC measurements provide base information to develop and track strategies aimed to reduce and

manage air pollution and climate change.

The understanding of the spatial and temporal variability of BC and its collateral effects, requires reliable, highly time-resolved, and long-term observations. To achieve this, three main aspects must be fulfilled during BC monitoring: (i) appropriate performance and quality check of the monitoring instruments, (ii) standardized use and maintenance by the operators, and (iii) reliable transmission and validation of data. The non-compliance of these requirements challenges the

accuracy and comparability of BC observations.

Defined as the most refractory portion of particles produced in combustion processes, with a strong light absorption capacity (Petzold et al., 2013), diverse techniques are available to measure black carbon in the atmosphere; depending on the measurement technique, BC may be addressed by different terminologies. When thermal methods are used, black carbon is measured as the non-volatized carbon remaining after applying specific high temperature to the sample, therefore BC is

called elemental carbon (EC). In laser-induced incandescence techniques, the sample is heated to vaporization temperatures using an infrared laser and the thermal radiation emitted by incandescent black carbon is measured, then converted to mass concentration; here, we measure BC as refractory black carbon (rBC). When optical methods are used, black carbon is called equivalent black carbon (eBC), because the mass concentration is indirectly retrieved from measurements of light attenuation (Bond et al., 2013; Lack et al., 2014). Other techniques used to measure black carbon include chemical oxidation and Raman

spectroscopy (Petzold et al., 2013).

In optical methods, the aerosol particle light absorption is measured either on the particles collected on a filter (filter-based absorption photometers), or measured directly in the aerosols suspended in a sample of air (photo-thermal spectrometers). In field monitoring, the filter-based absorption photometers (named in this document as FBAP) are widely used to perform long-term BC measurements, because these are robust, require relatively low human intervention, and no laboratory analysis

are needed to process the sample. The aethalometer (Hansen et al., 1982), an instrument quantifying the transmission of light through a filter where the aerosol particles are collected, is one of the commonly used FBAP instruments. The difference between the light from an internal source transmitted through the sample-laden filter relative to the clean part of the filter, is used to calculate the attenuation coefficient. The attenuation is transformed to absorption and later to eBC mass concentration using the black carbon mass absorption cross section and filter properties; these two lasts steps involve the use

of fixed correction factors and a compensation algorithm. Further description about the functioning of the instrument is given in section 2.1.

According to EBAS database (Tørseth et al., 2012), in the last 10 years (2011-2020) a total of 57 European stations or sites have reported data from particle light absorption measurements using filter-based photometers, including aethalometers. These measurements contribute to networks and projects such as ACTRIS (41 sites) and EMEP (5 sites) among others, and

some stations may contribute to more than one network at the time. Data from stations using FBAP in 29 non-European





countries are also available in EBAS. The COST Action CA16109 Chemical On-Line cOmpoSition and Source Apportionment of fine aerosoL COLOSSAL, reports in its catalogue the cooperation with 49 sites using FBAP in Europe.

Despite its wide use, the FBAP and particularly the aethalometers, feature inherent artifacts increasing the uncertainty in the measurements (Collaud Coen et al., 2010; Müller et al., 2011; Saturno et al., 2017). In first place, these instruments do

quantify directly neither the absorption nor the eBC mass concentration, these are instead estimated from the measurements of light attenuation caused by the aerosol particles. The absorption coefficients and concentrations are calculated based on different parametrizations and corrections for the absorption enhancement due to light scattering in the sample-laden filter matrix. In fact, the filter material used in the aethalometer and the particles immersed in it, scatter a portion of the incident light reducing the transmission of it through the filter (Weingartner et al., 2003). Therefore, a reduction in the light

transmitted may be taken as a higher absorption, with an additional small cross-sensitivity to scattering. A second artifact is caused by the loading effect produced by the aerosol particles accumulated in the filter matrix (Weingartner et al., 2003; Virkkula et al., 2007). After particles are deposited, the detection of changes in the attenuation decreases – saturates, causing an underestimation of black carbon absorption and in consequence, lower eBC concentrations (Drinovec et al., 2015; Drinovec et al., 2017).

The characterization of aethalometers is therefore required to understand and reduce the variability and uncertainty in the measurements of BC, and this can be done by comparison experiments (EEA, 2013; WMO, 2016). The intercomparison consists in placing two or more instruments to measure the same sample under equal conditions and time. By intercomparing, it is possible to study the instrument sensitivities to different aerosol sources and concentrations, the deviations caused by the type of filter material and numerical corrections used by the instruments, and the effects from

different operational and maintenance procedures.

One of the first documented aethalometers intercomparison was performed by Ruoss et al. (1993), contrasting ambient air measurements performed by the DLR aethalometer (DLR Research Centre) and the Hansen-type aethalometer (Magee Scientific); the authors found a significant up to 50 % variability among both instruments. In 1999, Hitzenberger et al. intercompared the absorption measurements carried out by an integrating sphere and one aethalometer (Hansen et al., 1984)

using different filter materials; the aethalometer underestimated the absorption coefficients in the range of -26 % to -66 % when using quartz fiber filters; while using glass fiber and membrane filters, the aethalometer overestimated the absorption up to 34 %. Hitzenberger et al. (2006) compared a wide range of optical and thermal methods measuring ambient air aerosols. The aethalometer AE9 reported higher eBC concentrations when compared against a multiangle absorption photometer (MAAP) (11.5 %) and a thermal-optical transmission (TOT) method (19 %); in the same study, when compared

against an integrating sphere, the aethalometer showed lower concentrations (-5 %). Collaud Coen et al. (2010) compared the absorption coefficients reported by aethalometers (AE10, AE16, and AE31) and MAAP in four different locations; depending on the algorithms used to correct the loading effect in the aethalometers, these instruments overestimated the absorption coefficients reported by the MAAP by 1 % to 33 %.



Although widely used, few experiments have been performed in order to characterize and compare the most recent
generation of filter-based absorption photometers used in BC monitoring (Drinovec et al., 2015). Extensive intercomparisons
are fundamental to determine the variability among instruments from stations supporting international collaborative projects.
They contribute to identify and quantify the factors influencing the instruments performance.

In this investigation, the authors present the results from the largest intercomparison of aethalometers model AE33, by the
characterization of 23 instruments. Three different aerosol sources were used in the workshops. The main goal is to
determine the unit-to-unit variabilities and their measurements throughout the spectral range covered by the AE33. Also, it
was studied the influence of the maintenance activities and accessories used by the instruments, in the reported eBC
concentrations. In the end, it is provided a series of recommendations for operation and maintenance.

## 2.  Materials and Methods

The intercomparison of aethalometers was conducted in three laboratory workshops carried out in the World Calibration
Centre for Aerosol Physics (WCCAP) in Leipzig, Germany. Seventeen AE33, part of the COST action CA16109
COLOSSAL and ACTRIS, were characterized during the first workshop performed from 14th to 25th January 2019 (Table 1).
Due to space limitations in the laboratory, the aethalometers were intercompared in four groups (A, B, C, D), completing 2.5
to 3 days of measurements by group. In the second workshop (group E, 7th to 12th June 2019), two aethalometers AE33 from
the Leibniz Institute for Tropospheric Research (TROPOS) were evaluated; both instruments monitored BC in a German
rural site as part of the one-year project "Zusatzbelastung aus Holzheizungen" (Additional Load from Wood Heating).
During the third workshop (group F, 18th to 20th June 2019), four aethalometers AE33 were intercompared. These four
instruments measured BC concentrations in Berlin, in the frame of the project "Orientierende Erfassung von BC in
Deutschland" (Orientated survey of BC in Germany).

### 2.1  The aethalometer AE33 and the compensation of eBC concentrations

The Aethalometer AE33 (Drinovec et al., 2015) uses a dual spot system to compensate the loading effect artifact. It
calculates the absorption and the compensated eBC concentrations from measurements of light transmission at seven
wavelengths from the near-UV to the near-IR (370, 470, 520, 590, 660, 880 and 950 nm).

The operation principle of the aethalometer consists in the continuous collection of aerosols on a filter, forming a sample-
laden spot. A light source illuminates the spot on the filter and, on the opposite side, a sensor measures the intensity of light
transmitted through it (signal $I$). The light transmission is also measured through a sample-free area on the filter and is used
as reference (signal $I_0$). By using both signals it is possible to calculate an attenuation (ATN; Eq. (1)).

$$ATN = -100 * \ln\left(\frac{I}{I_0}\right), \tag{1}$$



where the factor 100 is present for convenience only and ATN/100 should be used in further calculations. In the aethalometer, the change in the attenuation with time is assumed to be caused by the increase of black carbon mass deposited

on the filter. However, it was demonstrated that the correlation between the attenuation and the aerosol load on the filter is not linear at high attenuations (Gundel et al., 1984). Over time, the particles accumulated on the filter "shadow" each other, reducing the optical path length, saturating the signal and therefore the measurement of light transmission. This condition is known as the loading effect and causes an underestimation of eBC concentrations. It is a cumulative property that needs to be calculated in real time to accurately report eBC mass concentrations (Drinovec et al., 2015).

To overcome the loading effect, the AE33 collects the aerosol sample in two spots ($S_1$ and $S_2$) on the filter. Each spot is collected with different airflow rate, and the spot with higher flow becomes more loaded with sample (Fig. 1). The instrument measures the light transmission through both spots and calculates two attenuations ($ATN_1$ and $ATN_2$, using Eq. (1)), for the seven wavelengths of the instrument light source. The dual system allows to estimate a *compensation parameter* ($k$), based in the proportionality from the loading of both spots, to their airflows ($F_1$ from $S_1$ and $F_2$ from $S_2$), as shown in Eq.

180    (2),

$$\frac{F_2}{F_1} = \frac{\ln(1-k*ATN_2)}{\ln(1-k*ATN_1)}, \tag{2}$$

where the compensation parameter $k$, representing the loading effect, will be equivalent for both spots as they are loaded with the same sample of aerosols. Equation (2) is used to calculate the instantaneous compensation for each wavelength $k(\lambda)$.

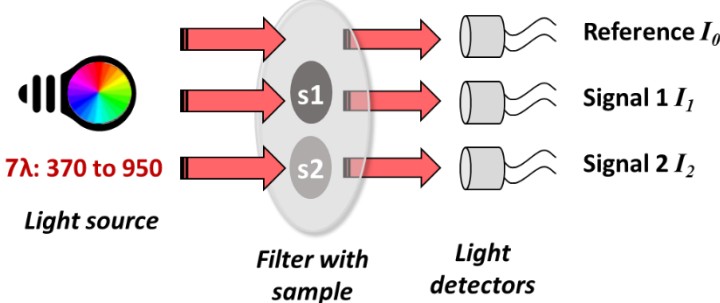

**Figure 1. Schematic representation of the optical chamber in the Aethalometer AE33.**

The intermediate step between ATN and eBC mass concentration is the absorption coefficient. Once the compensation is calculated, the $b_{abs}(\lambda)$ is estimated as shown in Eq. (3),

$$b_{abs}(\lambda) = \frac{s*(\Delta ATN_1(\lambda)\,/\,100)}{F_1(1-\zeta)*C*(1-k*ATN_1(\lambda))*\Delta t}, \tag{3}$$

where s is the spot area (constant, 0.785 cm$^2$), $F_1$ is the air flow through spot 1 (measured), $\zeta$ is the leakage factor (constant

adjustable, depends on the filter material and the leakage test), $ATN_1(\lambda)$ is the attenuation calculated for spot 1, $\Delta ATN_1(\lambda)$ is the change in the attenuation at each wavelength in a given time step $\Delta t$, $k$ the compensation parameter, and $C$ is the correction parameter for the multiple scattering enhancement. The particles and the filter may scatter a portion of light incident from the light source, increasing the optical path of the light in the filter and increasing the probability of light being



absorbed, i.e., a light absorption enhancement. In the AE33 the user set a constant value of $C$; there are specific values available for each type of filter tape (Magee Scientific, 2018); nevertheless, multiple studies have shown this $C$ factor depends also on the source of the aerosols measured (Collaud Coen et al., 2010; Ajtai et al., 2019), but this topic is out of the scope of this investigation.

Finally, the $b_{abs}$ and the BC mass absorption cross section ($\sigma_{air}(\lambda)$, fixed constants in the instrument), are used to calculate the eBC mass concentrations at the seven wavelengths, as shown in Eq. (4):

$$eBC(\lambda) = \frac{b_{abs}(\lambda)}{\sigma_{air}(\lambda)}, \tag{4}$$

### 2.2 Intercomparison procedure

The workshops were performed in three sessions (Fig. 2):

1. Initial comparison: during the first day the instruments were connected to the mixing chamber and started the measurements of urban background aerosols from Leipzig conserving the internal configuration and accessories provided by the operators. Only the instrument time was synchronized, and the measurement time resolution (1 min), the flow reporting standard (AMCA, 21.1 °C, 1013 hPa) and the maximum attenuation limit (ATN=120) were modified. After approximately 1 hour of ambient air measurements, synthetic soot produced with a miniCAST 5203 (Table 2) was supplied to the mixing chamber, followed by nigrosin particles (see section 2.4), particle-free air and ambient air, each one at a time, during 3 to 5 hours by sample. The initial comparison was performed with the aim to: (i) allow the adjustment of the internal compensation parameters $k(\lambda)$ to the local conditions and, (ii) determine the initial variability and deviation of the aethalometers before the maintenance and calibration procedures.

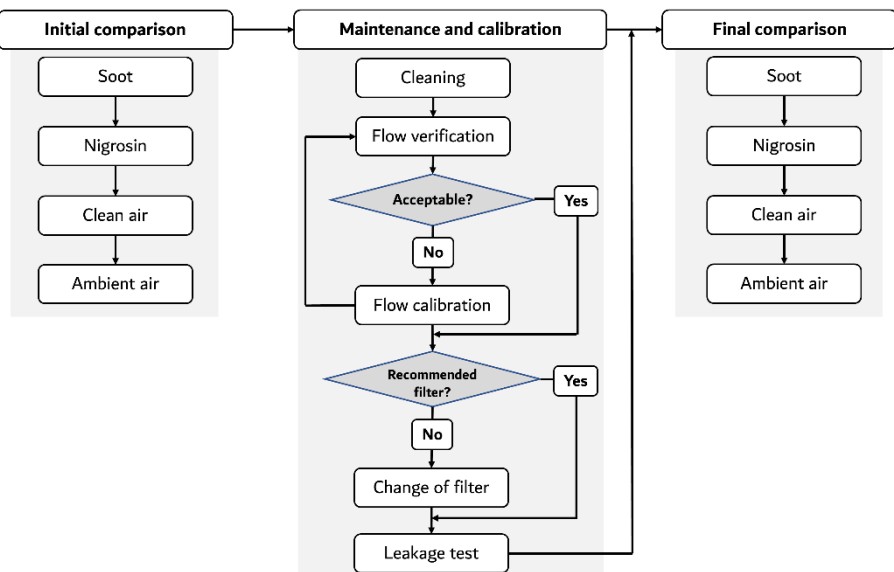

**Figure 2. The intercomparison procedure.**





2. Maintenance and calibration: the instruments were disconnected of the mixing chamber. The maintenance included a series of procedures performed by following the instructions given in the AE33 user manual - version 1.57 (Magee Scientific, 2018):

- Flow verification test, using an externally calibrated flowmeter (Table 2).
- Cleaning of the optical chamber.


- Flow calibration, performed only in those instruments with non-acceptable results from the flow verification test (deviations >10 %).
- Leakage test.
- Replacement of the filter tape, performed for instruments using a different filter tape to the one recommended currently (M8060).

3. Final comparison: the aethalometers measured the same three aerosol sources used in the initial comparison (synthetic soot, nigrosin particles, ambient), and particle-free air. The goal was to determine the new instrument-to-instrument variabilities after maintenance.

### 2.3 Experimental set-up

A mixing chamber (0.5 m$^3$) with an internal fan was used to distribute well-mixed samples of aerosols to the aethalometers (Fig. 3). The instruments intercompared measured eBC from ambient air, synthetic soot and nigrosin particles. The reference aethalometer from the WCCAP was also connected to the mixing chamber, measuring eBC aerosols in parallel with the aethalometers intercompared (Table 1).

     A Mobility Particle Size Spectrometer was used to quantify the particle number size distribution of the different aerosol

samples used during the workshops. Table 2 presents a list of the auxiliary instruments used during the intercomparisons.





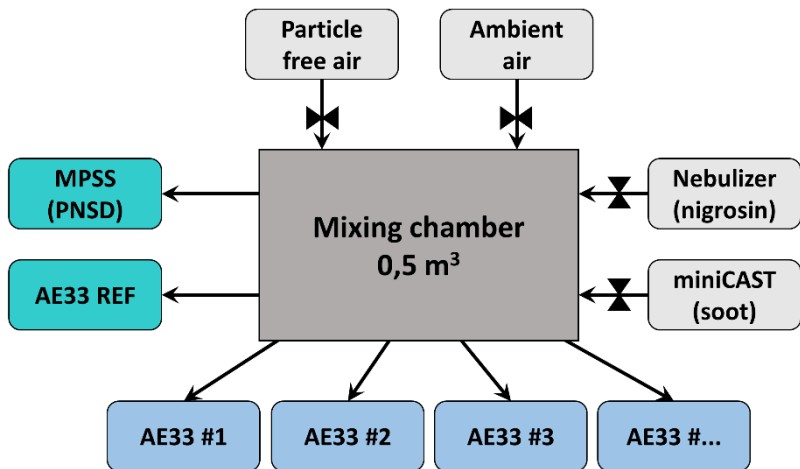

Figure 3. Experimental set-up used in the intercomparisons.





**Table 1. List of aethalometers AE33 intercompared.**

| Group | N° | ID | Serial | Intercomparison | | Operating conditions |
| --- | --- | --- | --- | --- | --- | --- |
| | | | | Date start | Date end | |
| A | 1 | A01 | S02-00246 | 14/01/2019 | 16/01/2019 | |
| | 2 | A02 | S07-00618 | | | |
| B | 3 | B01 | S02-00170 | 16/01/2019 | 18/01/2019 | |
| | 4 | B02 | S01-00080 | | | |
| | 5 | B03 | S07-00767 | | | |
| | 6 | B04 | S04-00387 | | | |
| | 7 | B05 | S0200267 | | | |
| | 8 | B06 | S02-00204 | | | |
| C | 9 | C01 | S01-00113 | 21/01/2019 | 23/01/2019 | Measurement time resolution: 1 min |
| | 10 | C02 | S01-00114 | | | ATN max: 120 |
| | 11 | C03 | S06-00560 | | | Filter tape: see table 3 |
| | 12 | C04 | S07-00729 | | | Inlet flow: 5 L min$^{-1}$ |
| D | 13 | D01 | S07-00669 | 23/01/2019 | 26/01/2018 | Flow reporting conditions: AMCA, 21 °C, 1013 hPa |
| | 14 | D02 | S00-00049 | | | |
| | 15 | D03 | S02-00258 | | | |
| | 16 | D04 | S00-00055 | | | |
| | 17 | D05 | S02-00156 | | | |
| E | 18 | E01 | S02-00202 | 07/06/2019 | 12/06/2019 | |
| | 19 | E02 | S07-00737 | | | |
| F | 20 | F01 | S07-00705 | 19/06/2019 | 21/06/2019 | |
| | 21 | F02 | S07-00706 | | | |
| | 22 | F03 | S05-00443 | | | |
| | 23 | F04 | S06-00578 | | | |





**Table 2. Instruments employed during the workshops.**

| Instrument | Measurement | Operating conditions |
|---|---|---|
| *Intercomparison* | | |
| **Aethalometer used as reference: model AE33, Magee Scientific (TROPOS), S/N: S02-00163** | eBC concentration at 7 wavelengths: 370, 470, 520, 590, 660, 880 & 950 nm | Measurement time resolution: 1 min<br>ATN max: 120<br>Filter tape: M8060<br>Inlet flow: 5 L min⁻¹<br>Flow reporting conditions: AMCA, 21.1 °C, 1013 hPa |
| **MPSS (Mobility particle size spectrometer): CPC model 3010 from TSI Inc. & DMA from TROPOS (WCCA reference)** | Particle number size distribution. Aerodynamic diameter range: 10 – 800 nm | Measurement time resolution: 5 min<br>Inlet flow: 1 L min⁻¹<br>Flow reporting conditions: Standard, 0 °C, 1013.25 hPa |
| **Soot Generator miniCAST: model: 5203 Type C, Jing Ltd.** | | Diffusion flame conditions:<br>• Propane: 105 mL min⁻¹;<br>• Oxidation air: 3.6 L min⁻¹;<br>• Dilution air: 20 L min⁻¹;<br>• Quench gas N₂: 20 L min⁻¹<br>Flow reporting conditions: Normal, 20 °C, 1013.25 hPa |
| **Customized particle nebulizer: built using a Constant Output Atomizer model 3076, TSI Inc.** | | Nigrosin:<br>• CAS: 8005-03-6<br>• Molecular weight: 202.2 g mol⁻¹<br>• Concentration of the solution: 0.5-0.8 g L⁻¹ |
| *Maintenance* | | |
| **Mass flow meter: model 4140 F, TSI Inc.** | | Measurement time resolution: 1 sec<br>Operative range: 0.01 – 20 L min⁻¹<br>Flow reporting conditions: AMCA, 21.1 °C, 1013 hPa |

## 2.4 Aerosol sources

The aethalometers measured eBC concentrations from three aerosol sources:



1. Synthetic soot particles produced with a miniCAST (Jing Ltd, 2013), using a fuel-lean mixture (fuel-to-air equivalence ration, φ < 1). Table 2 shows the operating conditions used in the miniCAST during the intercomparisons.

2. Black particles created by the nebulization of a Nigrosin solution (Table 2).

3. Ambient air aerosols from the urban background in the city of Leipzig, Germany. Concentrations correspond to early-morning periods (3:00 to 9:00 am), during winter (workshop 1) and summer time (workshop 2 and 3).

Figure 4 presents the average particle number size distributions of the aerosol sources measured during the workshops.

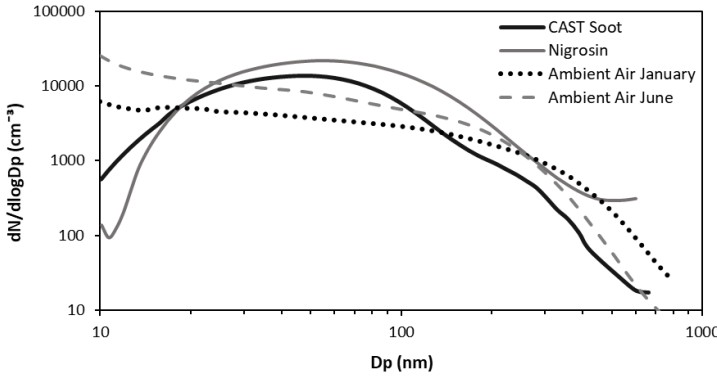

**Figure 4. Particle number size distributions of the aerosol sources used in the intercomparison.**

### 2.5 Data processing and analysis

eBC concentrations were measured every minute. Subsequent to the workshops, the data from the aethalometers were cleaned based on the instrument status codes: the AE33 reports a series of status codes representing the operational state, internal procedures in curse, warning alerts or errors in the instrument, e.g., 0: normal measurements, 1: filter tape advance,
8: Check flow status history, 384: tape error (tape not moving, end of tape) (Magee Scientific, 2018). As more than one condition may occur at the same time, these statuses are built under a binary system therefore they may be numerically combined, e.g., 9: Check flow status history (warning) + Tape advance (procedure), 387: Tape error (tape not moving, end of tape) + Stopped. Therefore, data from the few valid statuses available were kept and then used in the data analysis:

0: Measurement;
8: Check flow status history (warning);

128: Tape warning (less than 30 spots left);

136: Tape warning (less than 30 spots left) + check flow status history;

256: Tape last warning (less than 5 spots left);


65535: Database bigger than $2*10^6$ lines, i.e., a warning because the memory of the instrument is getting full - the user
has to make sure the data are recorded correctly, in some cases data with this status are not valid because overwriting
conflicts.

To compare the measurements made by the aethalometers and the reference AE33, we used Deming total least squares
regression analysis (R package "Deming" (Therneau, 2018)), to account for the independent observational errors from each
instrument and the reference (dependent and independent variables). Deming regression finds the best fitting line by
minimizing the sum of the distances in both x and y directions, simultaneously (Cornbleet and Gochman, 1979).

The processes of data cleaning and analysis were performed in the software R studio version 1.2.1335-1.

## 3.   Results and Discussion

The analysis in section 3.1 is a detailed analysis of the instruments characterized in the group D, as a case of study
illustrating diverse situations observed during the workshops. A summary and analysis of the results of the total of 23
intercompared units is given in section 3.2.

### 3.1  Unit-to-unit variability, case of study group D

The experiment in group D was divided in three sections: an initial comparison before maintenance, an intermediate
comparison after a partial maintenance, and a final comparison after filter tape change.

Initial comparison
Figure 5 presents the time series of the one-minute eBC mass concentrations measured in the initial comparison by the
aethalometers within group D. The gray areas represent the periods when the different aerosol sources and particle free air
were supplied in to the mixing chamber; red bars indicate times when tape advances occurred in more than one instrument at
the same time.
The variability in the measurements of eBC observed in group D, and in general in the six groups, were significantly higher
few minutes before and after a tape advance (TA). Continuous supply of soot with concentrations above 15 to 20 µg·m$^{-3}$ led
to the instrument reaching the maximum attenuation limit (ATN$_{TA}$=120) after ~30 min, inducing a TA in the majority of the
instruments. The differences in the reported eBC mass concentrations close the TAs reach up to 25% among the AE33.
During nigrosin measurements, constant concentrations of 10 µg·m$^{-3}$ led to the maximum attenuation limit after ~120 min.
Offsets in the concentrations measured by the aethalometers were up to 25% during the nigrosin supply.
Before the maintenance and calibration interventions, the highest deviations in group D were seen in three aethalometers
(Fig. 5): D03, D04 and D05. The instruments D04 and D05 underestimated the eBC concentrations by 11% and 25% with





respect to the reference, respectively. These two instruments used an older version of filter tape M8050 (also known as TX40), a glass fiber filter on a woven backing. The M8050 filter tape was distributed during a short period from 2016 to 2017, and according to the manufacturer it was substituted because of evidence of unsatisfactory performance. On the other hand, the instrument D03 reported slightly higher concentrations than our reference aethalometer, overestimating the eBC concentrations by up to 6 and 8%, while measuring soot and nigrosin particles, respectively. The aethalometer D03 used the

T60A20 filter tape (also known as M8020 or AE33-FT), made from TFE-coated glass fibers, which was the first filter used in the AE33 and has been since discontinued.

The different materials used in, and the structure of the filters, give them specific light scattering properties responsible of light absorption enhancement (Petzold et al., 1997). Therefore, the correction factor $C$ accounting for the light scattering effects of the filter and the particles immersed in it, take different values for each type of filter tape. It has been demonstrated

that the apparent $C$ correction factor is in addition susceptible to the type of aerosols measured (Collaud Coen et al., 2010). However, the estimation of a source-dependent $C$ factor was not in the scope of this study; our interest is limited to the correct use of the $C$ factors associated to each filter tape. Standard $C$ factor for the filters previously available, M8050 and T60A20, is 1.57, relative to the value 2.14 determined in Weingartner et al. (2003). To guarantee the comparability within monitoring stations, these two filter tapes are no longer recommended to be used in the AE33 (Magee Scientific, 2018). The

new filter M8060 must be used instead, and its corresponding multiple scattering parameter $C$ has to be set as the internal parameter of the instrument ($C_{M8060} = 1.39$).

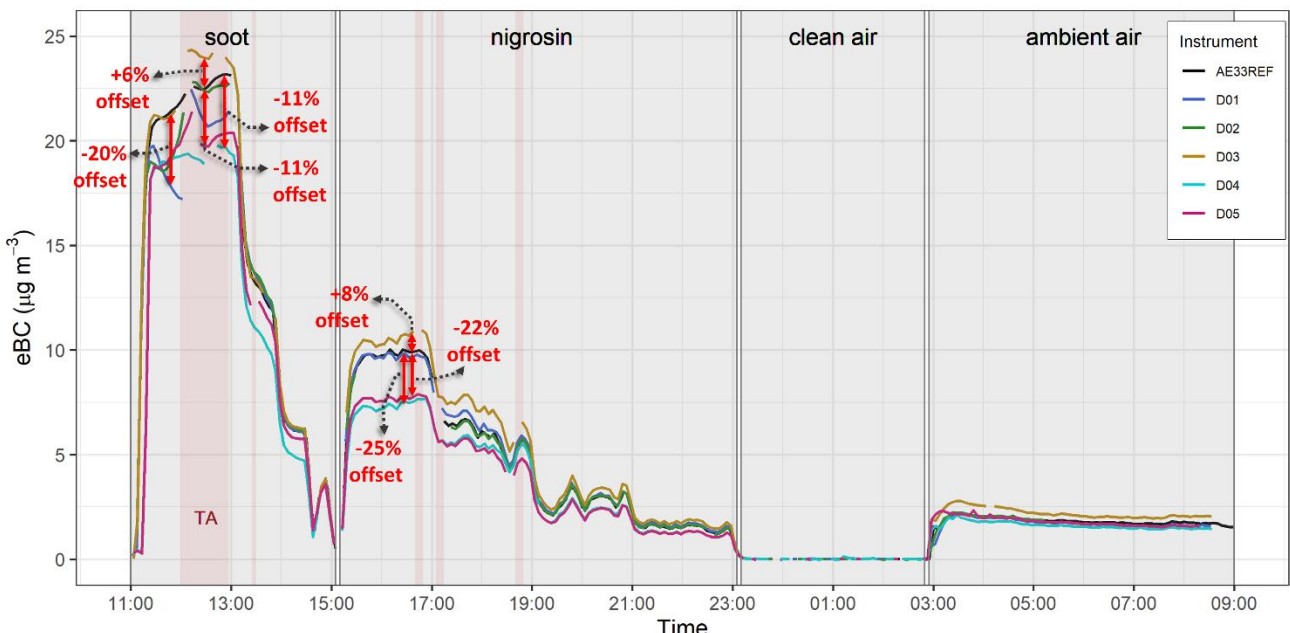

**Figure 5. Time series of eBC mass concentrations at 880 nm before maintenance in group D.** Gray panes show the periods of clean air and aerosols supply, red bars indicate times when tape advances occurred in more than one instrument simultaneously.





General maintenance and intermediate comparison

In group D the maintenance was divided in two phases to observe separately the influence of the essential servicing activities, i.e., cleaning of the optical chamber, flow verification and calibration, and leakage test, versus the second phase – the replacement of the older filter tapes.

During the first phase of maintenance, the instrument D03 showed unsatisfactory results from the flow verification and

leakage tests. In average, the flow sensors detected a 30% less airflow than the reference flowmeter (see Table 2), requiring a flow calibration. In the aethalometer AE33, if the flow verification test indicates a deviation of ±10% in any of the three flow rates (flow through spot 1, spot 2 and common flow), a flow calibration must be carried out. From the leakage this instrument reported a leakage of 9%. This result indicates that almost a ten percent of the inlet air flow in the aethalometer ($F_{in}$), is being tangentially lost across the edges of the filter tape. The results from the flow verification and leakage tests for

the other instruments in group D were satisfactory (Table 2).

During the experiment after partial maintenance (intermediate comparison), the instrument D03 underestimated by 29% and 18% of the eBC concentrations during soot and nigrosin measurements, respectively (Fig. 6a). The instrument reported an overestimation of the eBC mass concentrations, in contrast with the underestimation in the initial comparison. During the same period, the instruments D04 and D05 continued underestimating the eBC concentrations while measuring soot (21%)

and nigrosin (27%).

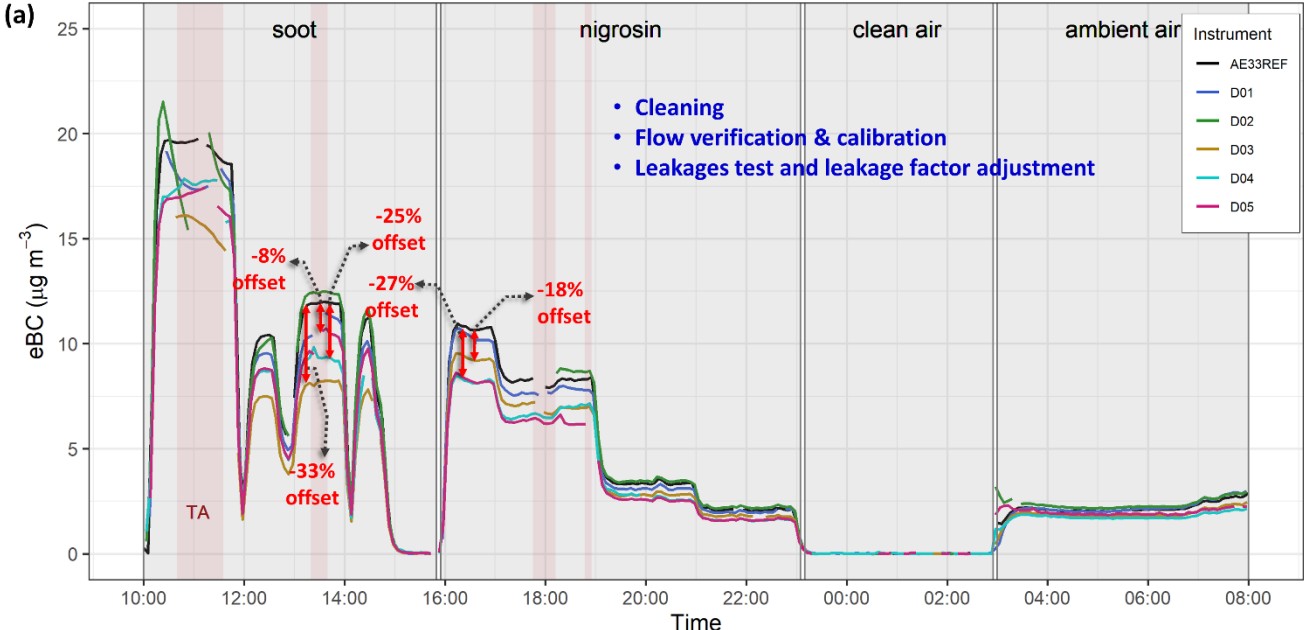



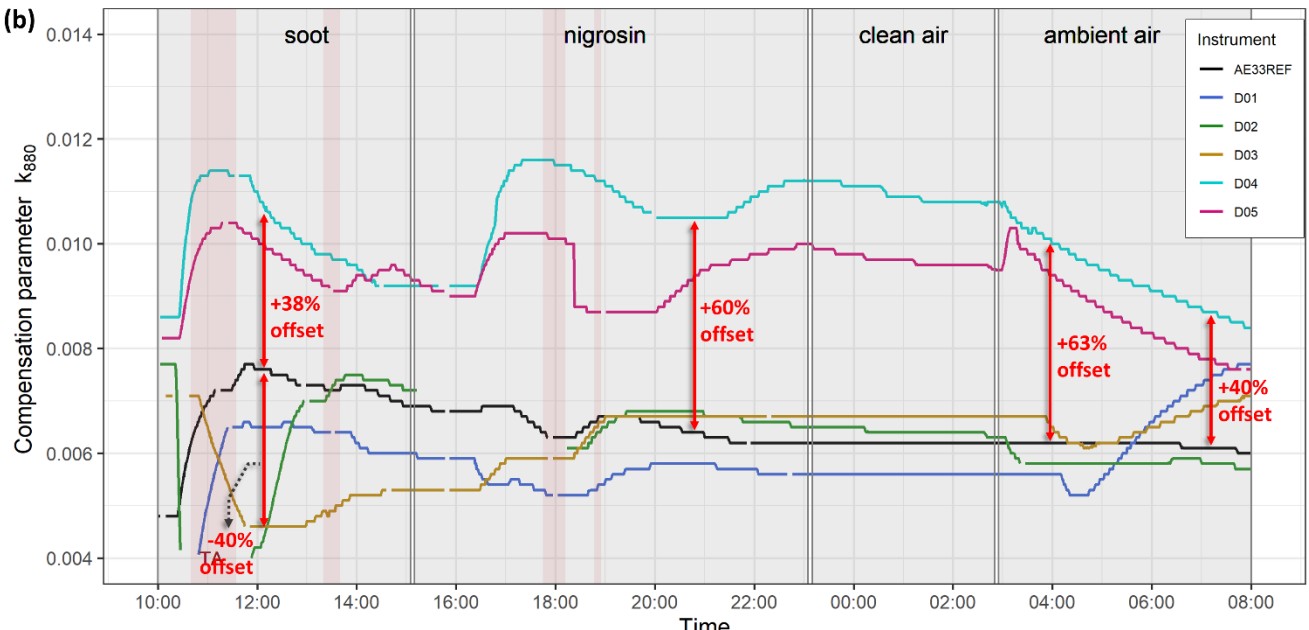

**Figure 6. Time series of (a) eBC mass concentrations at 880 nm and (b) compensation parameters $k_{880}$, after the initial phase of maintenance in group D.** Gray panes show the periods of clean air and aerosols supply, red bars indicate times when tape advances occurred in more than one instrument simultaneously.

The shape of the sample spots also demonstrates problems in an aethalometer. This was evident in the instrument D03 whose sample spots presented an irregular shape with heterogeneous saturations/streaks (Fig. 7a); under optimal operating conditions the spots formed on the filter tape have a circular and well-defined shape, homogeneously filled with sample which assures a correct measurement of the attenuation change (Fig. 7b). However, it was determined that the streaks (Fig. 7a) do not impact the measurement (John Ogren, personal communication).

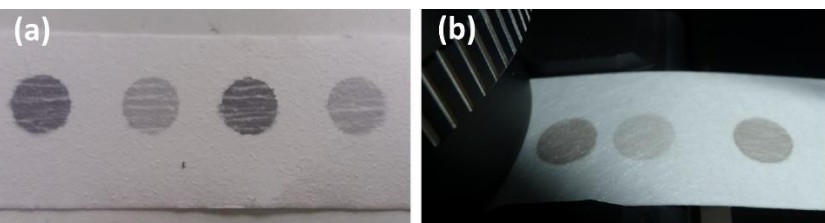

**Figure 7. Shapes of sample spots observed during maintenance in group D. (a) instrument D03, (b) instrument D01. These instruments used different tape material.**

The real time calculation of the eBC mass concentrations in the AE33 is based in the compensation parameters $k$. Figure 6b illustrates the time series of the $k_{880}$ values during the intermediate comparison (after flow and leakage adjustments only). The comparison of the compensation parameters is significant as these respond to the changes of aerosol sources and





concentrations in the experiments, independently of instrument correction factors. Once the flow verification and calibration
were implemented, it would be expected to have similar $k$ values among the instruments. Nevertheless, as evidenced in
Figure 6b, the differences in the $k$ from the reference aethalometer and the instruments in group D ranged from -40 % until >
+60 %. The main reasons explaining these deviations are the time at which the TAs occurred in the instruments, and the
filter materials they used. Although the instantaneous $k$ values are calculated using Eq. (2), the instrument uses specific $k$
depending on a weighting method based on the values of the attenuation as follows (supplementary material from Drinovec
et al., 2015):

- **$ATN_1 < 30$**: Under low BC concentrations as well as after one tape advance, a low attenuation change will take
  place in the aethalometers. The extent of $ATN_1 < 30$ (attenuation from in spot 1, channel 1), depends on the black
  carbon concentrations and the sampling time resolution. Under these conditions, the last $k$ values from the previous
spot ($k_{old}$), will be used to compensate the eBC mass concentrations. Accordingly,
  $$k = k_{old} \qquad (5)$$
- **$30 < ATN_1 < 120$**: For this attenuation range, the $k$ is calculated using the $k_{old}$, and the instantaneous $k$ calculated
  from Eq. (2):
  $$k = k_{weighted} = \frac{(ATN_{TA} - ATN_1)*k_{old} + (ATN_1 - ATN_{f2})*k_{inst}}{(ATN_{TA} - ATN_{f2})}, \qquad (6)$$
where $ATN_{TA}$ is the maximum limit attenuation triggering a tape advance and $ATN_{f2}$ is the upper limit attenuation for
  the fitting range, $ATN_{f2} = 30$.
- **$ATN_1 = 120 = ATN_{TA}$**: Once the spot 1 is completely loaded and the threshold attenuation is reached, the $k$
  becomes equals to the instantaneous $k$ calculated with Eq. (2).

The given $k$ values also depend on the filter type as the different materials determine the filter loading rate, consequently the
moment when the threshold attenuation ($ATN_{TA}$) is attained. In addition, the $k$ values are susceptible to the type of aerosols
measured (composition and size) and their mixing state (Drinovec et al., 2017). From the mathematical definition (Eq. (3)
and Eq. (4)) the $k$ values are inversely proportional to eBC, as observed in the instruments D04 and D05 with the higher
positive deviations from our reference aethalometer (Fig. 6b). Perhaps during soot and nigrosin measurements, the $k_{880}$ for
these two instruments followed the same trend as the reference AE33, a drastic change occurred during ambient air
measurements. Through this last period, the $k$ from both instruments decreased constantly, meaning this couple of
aethalometers detected a different attenuation range (between 30 and 120), instead of a low attenuation associated to low
ambient air concentrations taking place. Along the intermediate comparison the compensation parameters exhibited a
significant variability in group D, ranging from 0.0045 to 0.0115. When performing an intercomparison, the change of the
sample requires a "dry run" of a sample spot up to $ATN_{TA}$ to obtain the source specific value of the parameter $k$.
Alternatively, all data needs to be manually reprocessed.


Final maintenance and comparison

In the final step of maintenance, the filter tape was replaced in three instruments from the group D: D03, D04 and D05,
which were using older versions of filter tapes which are no longer recommended for the AE33. The final comparison was
performed during two days; as observed in Figure 8a, the deviations among the eBC mass concentrations reported by the
instruments reduced significantly for all aerosol sources (<10 %), in comparison with the initial and intermediate
comparisons performed in group D. Figure 8b shows the time series of the compensation parameters $k$ at 880 nm
corresponding to the final comparison. The new $k$ ranged now from 0.005 to 0.008, and their deviations from the reference
aethalometer ranged from -20 % to -10 % during soot and nigrosin. However, substantial differences took place during
ambient air measurements mainly in the aethalometer D05; once the supply in the mixing chamber changed from clean air to
ambient air at midnight, the compensation parameters from D05 jumped abruptly between > 0.012 to negative values within
1 hour. This situation is associated with the response of the instrument to changes in pressure: the airflow pressure in the
mixing chamber affects the flow rates, which is directly related with the instantaneous $k$. During maintenance, the
400    aethalometer D05 presented unusual behavior associated with the flow ratio appearing in a certain extent even after flow
calibration. As recommended by the manufacturer, the ratio of $F_1$ (higher sample airflow through spot 1) to $F_2$ (lower sample
airflow through spot 2), should be about 1.75 to 2.5, however this relation was not always accomplished in this instrument.
As result, it was recommended to the operator to send the instrument to the manufacturer for maintenance.

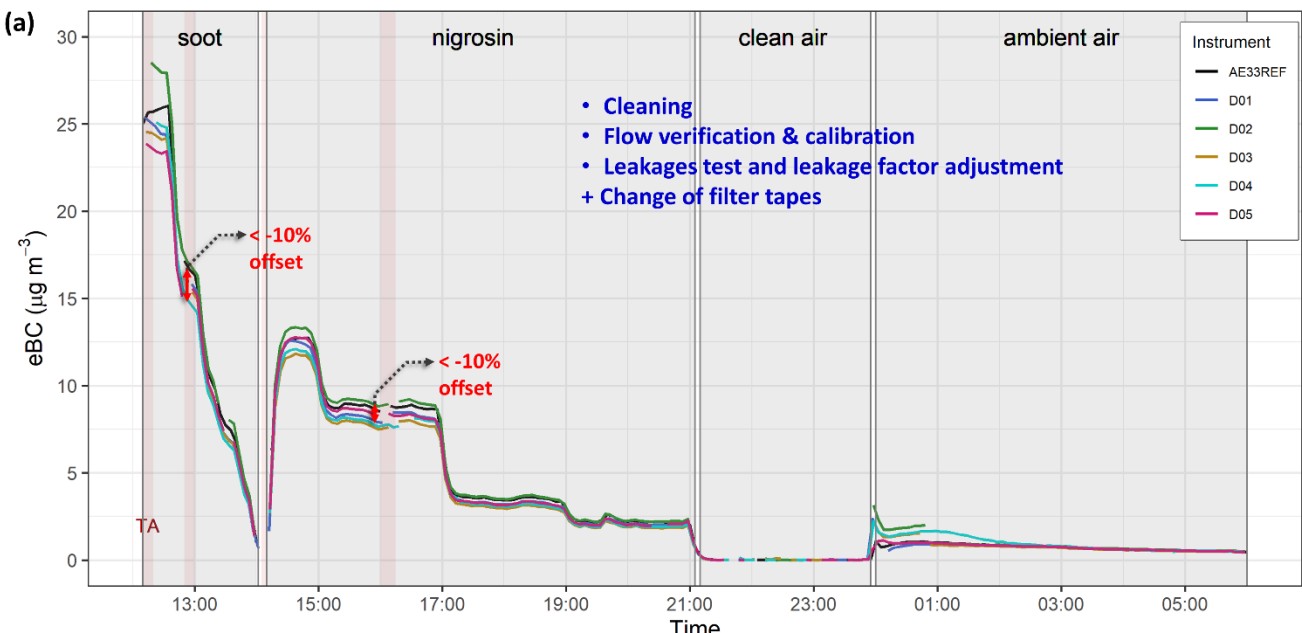





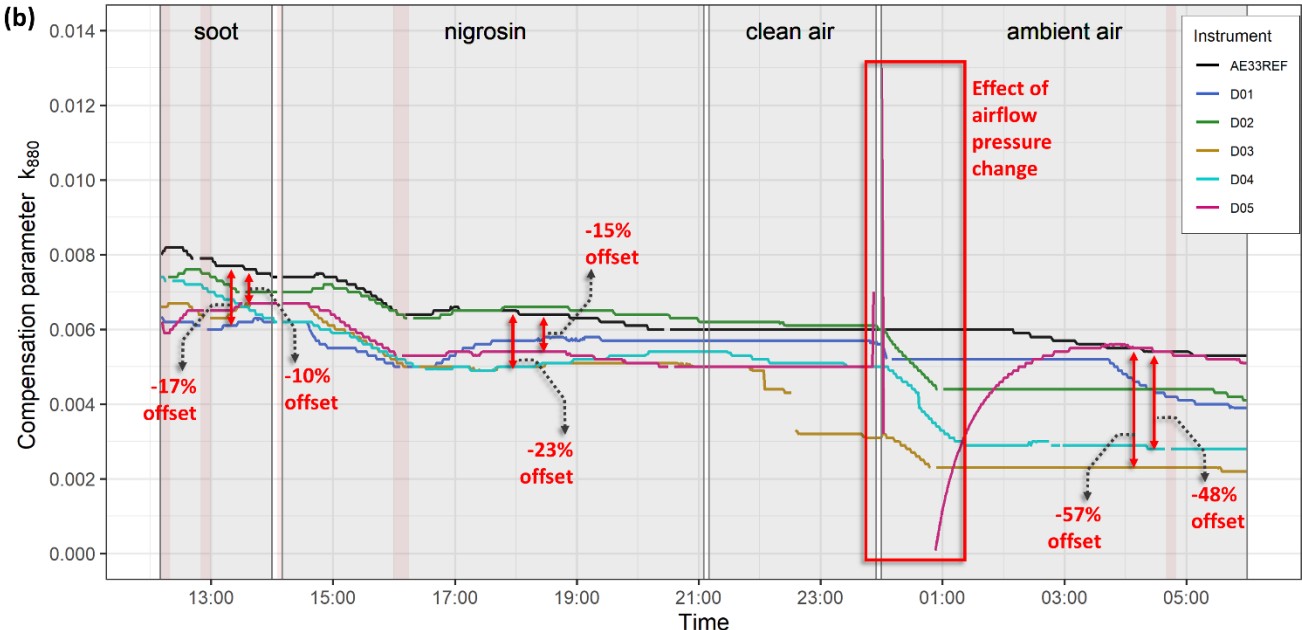

**Figure 8. Time series of (a) eBC mass concentrations at 880 nm and (b) compensation parameters $k_{880}$, after the final phase of maintenance in group D.** Gray panes show the periods of clean air and aerosols supply, red bars indicate times when tape advances occurred in more than one instrument simultaneously.

Statistical summary

The numerical unit-to-unit variabilities were calculated using data from measurements at 880 nm before and after maintenance, via total least squares regression with intercepts forced to zero. In the case of group D, after maintenance includes flow, leakages and filter tape adjustments.

Figures 9, 10 and 11, present the scatterplots from the comparison of the five instruments in group D against our reference aethalometer, while measuring soot, nigrosin and ambient air, respectively. For soot and nigrosin, the correlations were built using measurements performed after one or two tape advances, to avoid the bias caused by the effects previously described in this section. In the case of ambient air, the measurements were averaged to 5 minutes as the ambient concentrations of BC were low (< 1 µg m$^{-3}$) after maintenance.

The results from the regressions demonstrate acceptable agreement between the aethalometers in group D, in general improved after maintenance activities. In soot measurements, the total average slope ranged from 0.950 (5 % deviation) before service, to 0.965 (3.5 % deviation) after service. In the case of nigrosin, the total average slope remained almost constant ranging from 0.947 (5.3 % deviation) before service, to 0.944 (5.6 % deviation) after service. From the ambient air observations, low variabilities were estimated in group D however, it is inaccurate to state an improvement or not regarding the measurements before and after maintenance, as these concentrations were very low and rather stable, making the interpretation of the regression imprecise. In ambient air measurements, the instrumental noise (described in section 3.3),





becomes critical in sites with low BC concentrations. Before maintenance in group D, the one-minute average eBC concentration measured at 880 nm by the reference aethalometer in the early morning time, was 1.82 µg m$^{-3}$, and the average noise calculated in this group was 0.038 µg m$^{-3}$, representing 2% of the reported eBC concentration. After maintenance, the average eBC in the site was 0.65 µg m$^{-3}$ (880 nm), and the average noise calculated was 0.037 µg m$^{-3}$, representing 6 % of the average eBC.


Table 3 presents a summary for the regression analysis in group D and the other 18 aethalometers intercompared: the relative slope (AE33 vs. AE33 REF) and determination coefficient $r^2$ before and after service.

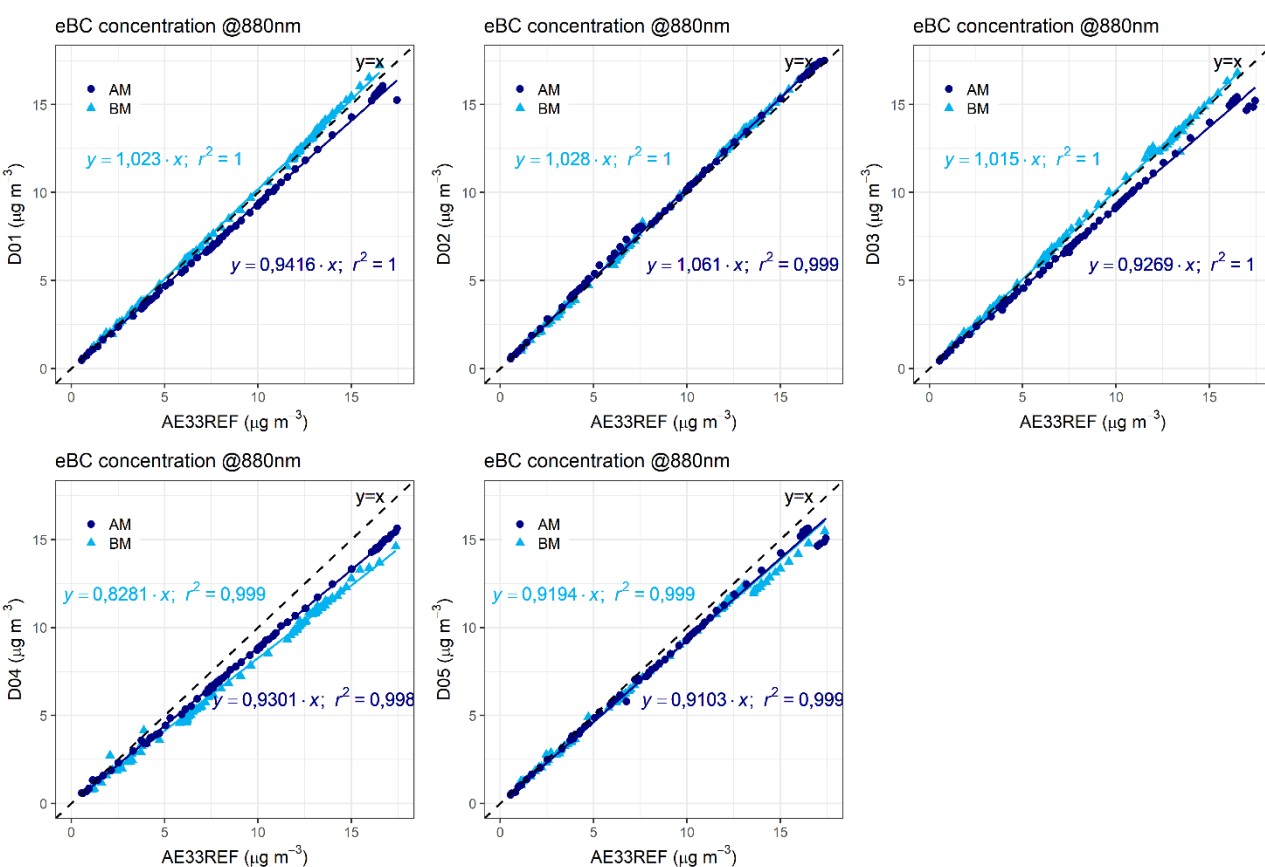

**Figure 9. Regression for the comparison of the instruments in group D and the reference aethalometer, before and after maintenance during soot measurements.** BM: before maintenance, AM: after maintenance.





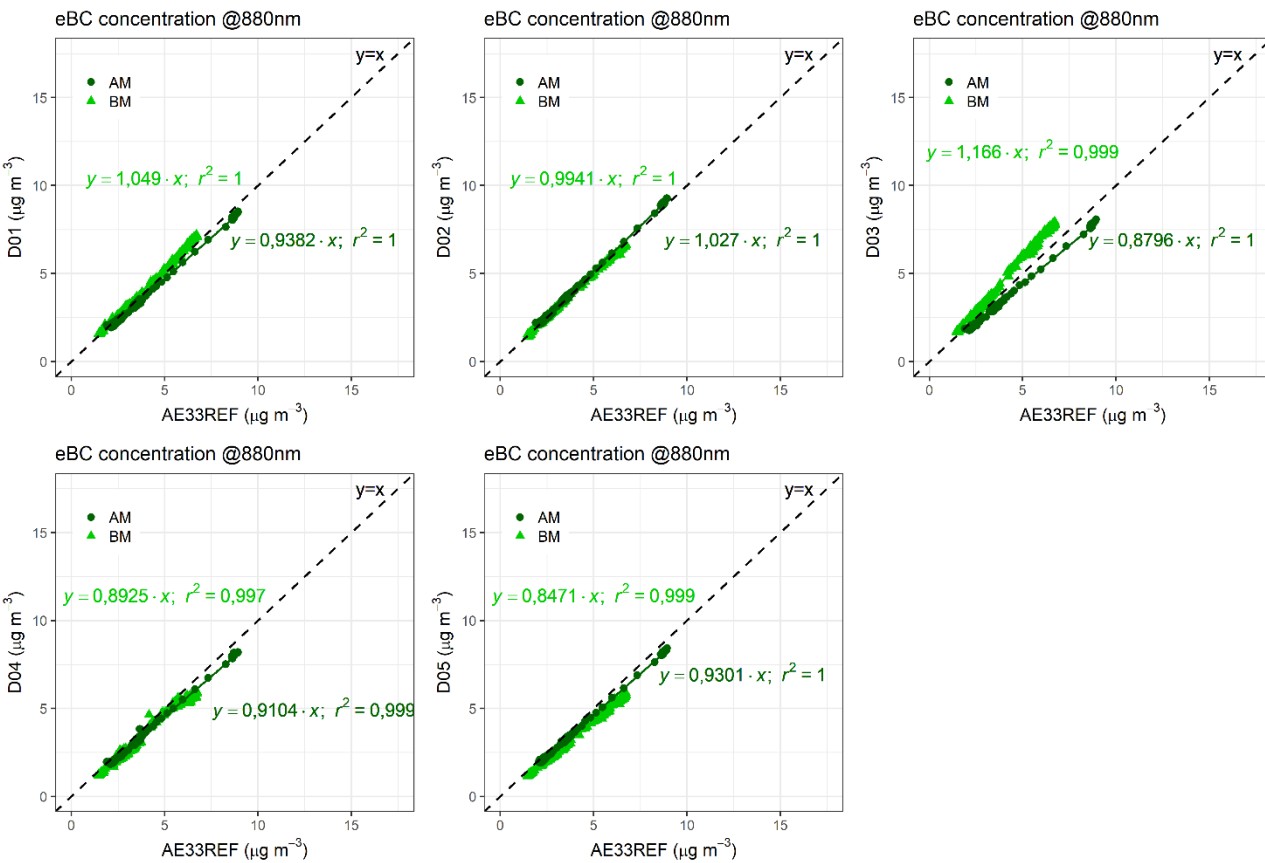

**Figure 10. Regression for the comparison of the instruments in group D and the reference aethalometer, before and after maintenance during nigrosin measurements.** BM: before maintenance. AM: after maintenance.





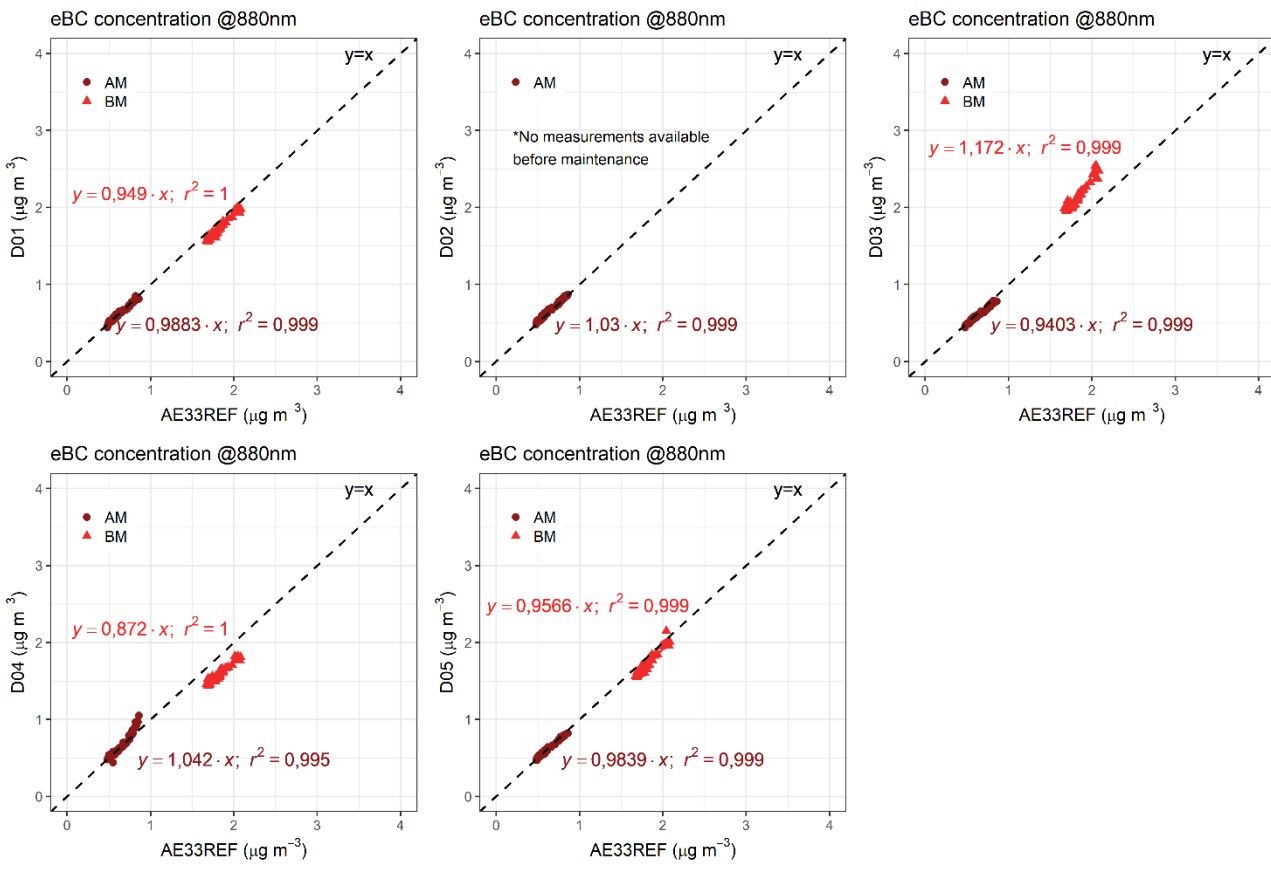

**Figure 11. Regression for the comparison of the instruments in group D and the reference aethalometer, before and after maintenance during ambient air measurements.** BM: before maintenance. AM: after maintenance.

### 3.2 Total unit-to-unit variability of the 23 aethalometers

In general, the aethalometers in groups A to F showed acceptable agreement when compared against our reference aethalometer, and this improved in most of the cases after the maintenance activities. Out of the 23 aethalometers intercompared, five instruments from groups B and D exhibited the highest unit-to-unit deviation. Regarding the eBC concentrations (880 nm), the total average deviation from the 23 instruments was 1.1 % before maintenance and 0.4 % after maintenance, for soot measurements. In nigrosin, the total average deviation changed from -0.6 % to -1.1 % before and after maintenance, respectively. A fair comparison for ambient air measurements is not possible to calculate, considering the significantly low and stable concentrations measured during some days in the urban background in Leipzig, and the fluctuations of concentrations in the workshops. The deviations calculated for the 23 aethalometers are summarized in Table 3.

Flow verification and leakages tests results were acceptable for most of the 23 instruments and are listed in Table 2; both results from the flow verification before and after maintenance are shown only for those instruments whose initial results





were unsatisfactory and whose tests had to be repeated after maintenance. The flow calibration procedure was applied to five instruments; these presented initially higher deviation in the lower flow tested (1 L min$^{-1}$; 16 % in average), followed by the high flow (3 L min$^{-1}$, 9 % in average) and the total flow (5 L min$^{-1}$; 7 % in average). After maintenance the new results from the flow verification tests showed average deviations of 1.8 %, 0.9 % and 0.1 %, for the airflows of 1, 3 and 5 L min$^{-1}$, respectively.

From the six groups, only three required a change of filter tape: D03, D04 and D05 (Table 2).

Wavelength-dependency of the unit-to-unit variability

The change in the unit-to-unit variabilities according to wavelength was analyzed, by calculating total least squares regressions of BC concentrations measured at the seven channels of the twenty-three aethalometers intercompared. The

influence of the maintenance in these variabilities was also investigated. Figure 12 shows the boxplots representing the range of the deviations calculated for soot, nigrosin and ambient air. Slope equaling to 1 (red line) indicates the instruments are 1:1 in respect with the reference (0 % of deviation).

As seen in the figure, no significant changes in the variabilities throughout the spectral range was seen during soot measurements, but slightly negative deviations were observed at 660 nm. The total range of variabilities also had a slight

reduction after maintenance and the median slope values were closer to 1 during this period. No significant changes were observed with wavelength for nigrosin measurements; however, as mentioned earlier, the total average variability increased after maintenance. In both measurements, before and after maintenance, the median deviations were lower than ± 5 % (slopes ranging from -0.97 to 1.04) for nigrosin supply. Finally, the variabilities from ambient air showed a reduced range after maintenance. No clear trend was observed with wavelengths, but slightly more negative deviations were seen at 370

and 660 nm.





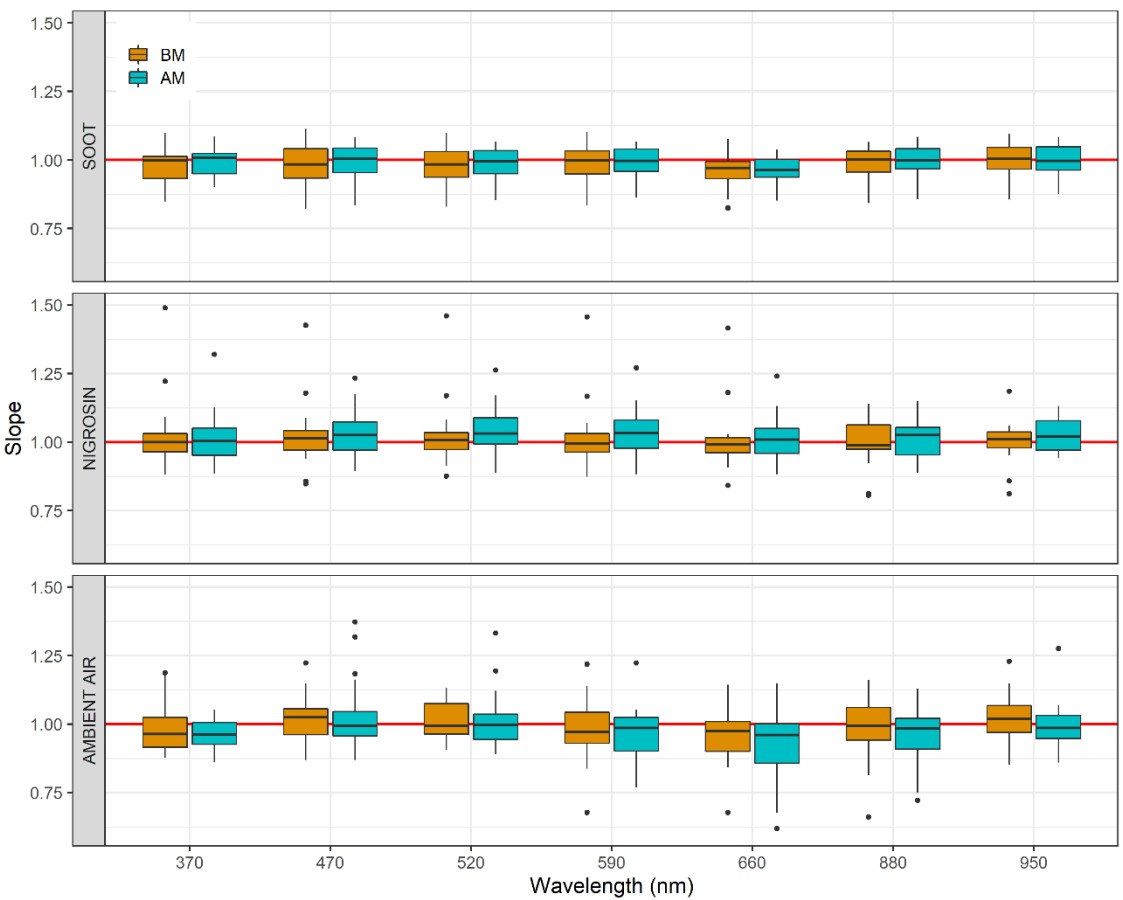

**Figure 12. Average unit-to-unit variabilities as function of wavelength for soot, nigrosin and ambient air.** The black horizontal line inside the boxes represents the median slope value; the lower and upper borders of the boxes are the first and third quartiles, on which the middle 50% of the slope values are located; the whiskers correspond to ranges for the bottom 25% and the top 25% of the slope values without outliers, which are represented by the black points. The red line represents the slope equaling to 1 (0 % of deviation).





**Table 3. Summary of results from maintenance tests.**

| Group | Instrument | BM or AM | Flow verification test[1] (%) F1 1 (L min⁻¹) | F1 3 (L min⁻¹) | F1 5 (L min⁻¹) | FC 1 (L min⁻¹) | FC 3 (L min⁻¹) | FC 5 (L min⁻¹) | Flow calibration | Leakage test (%) BM | Leakage test (%) AM | Initial type of filter tape | Change of filter tape |
|---|---|---|---|---|---|---|---|---|---|---|---|---|---|
| A | A01 | BM | 81 | 98 | 99 | 84 | 99 | 99 | ✓ | 2 | - | M8060 | - |
| | | AM | 104 | 101 | 101 | 102 | 101 | 101 | | | | | |
| | A02 | BM | 101 | 101 | 101 | 99 | 101 | 101 | - | 1.4 | - | M8060 | - |
| B | B01 | BM | 84 | 90 | 94 | 80 | 89 | 93 | ✓ | 1.9 | - | M8060 | - |
| | | AM | 99 | 98 | 99 | 95 | 97 | 98 | | | | | |
| | B02 | BM | 98 | 100 | 101 | 95 | 100 | 100 | - | 1.8 | - | M8060 | - |
| | B03 | BM | 99 | 101 | 102 | 97 | 101 | 102 | - | 2.4 | - | M8060 | - |
| | B04 | BM | 94 | 99 | 100 | 98 | 100 | 101 | - | 1.6 | - | M8060 | - |
| | | AM | 93 | 98 | 100 | 93 | 98 | 100 | | | | | |
| | B05 | BM | 82 | 93 | 95 | 80 | 92 | 95 | - | 1.9 | - | M8060 | - |
| | | AM | 95 | 99 | 101 | 91 | 98 | 100 | | | | | |
| | B06 | BM | 88 | 89 | 89 | 81 | 87 | 89 | ✓ | 6.5 | ND | T60A20[2] | - |
| | | AM | 101 | 100 | 100 | 101 | 100 | 100 | | | | | |
| C | C01 | BM | 99 | 102 | 103 | 101 | 103 | 103 | - | 1.9 | - | M8060 | - |
| | C02 | BM | 97 | 98 | 99 | 95 | 98 | 100 | - | 1.7 | - | M8060 | - |
| | C03 | BM | 101 | 100 | 100 | 97 | 99 | 100 | - | 1.8 | - | M8060 | - |
| | C04 | BM | 93 | 100 | 101 | 90 | 100 | 101 | ✓ | ND | 1.8 | M8060 | - |
| | | AM | 98 | 99 | 100 | 99 | 99 | 100 | | | | | |
| D | D01 | BM | 103 | 101 | 101 | 100 | 101 | 102 | - | 1.9 | - | M8060 | - |
| | D02 | BM | 98 | 99 | 99 | 101 | 99 | 99 | - | 8.4 | - | M8060 | - |
| | D03 | BM | 68 | 70 | 72 | 69 | 70 | 72 | ✓ | 9.2 | 1.5 | T60A20 | ✓ |
| | | AM | 102 | 100 | 100 | 102 | 100 | 100 | | | | | |
| | D04 | BM | 103 | 101 | 102 | 102 | 100 | 101 | - | 0.9 | - | M8050 | ✓ |
| | D05 | BM | 101 | 100 | 101 | 98 | 100 | 101 | - | 0.7 | - | M8050 | ✓ |
| E | E01 | BM | 98 | 99 | 100 | 99 | 99 | 100 | - | 1.8 | - | M8060 | - |
| | E02 | BM | 100 | 100 | 100 | 101 | 100 | 100 | - | 2.7 | - | M8060 | - |
| F | F01 | BM | 97 | 100 | 100 | 97 | 99 | 100 | - | 1.8 | - | M8060 | - |
| | F02 | BM | 96 | 101 | 101 | 96 | 101 | 101 | - | 1.7 | - | M8060 | - |
| | F03 | BM | 97 | 101 | 101 | 97 | 102 | 101 | - | 2 | - | M8060 | - |
| | F04 | BM | 101 | 101 | 101 | 99 | 101 | 101 | - | 2 | - | M8060 | - |

[1]Flow reporting standard: AMCA (21 °C, 1013 hPa). [2]This filter tape was not replaced because of operational reasons.

BM: Before maintenance. AM: After maintenance. ND: No data.





**Table 4. Relative slope and correlation coefficients for total least squares regression forced through the origin, for eBC mass concentrations (880 nm), before and after maintenance.**

| Instrument | Aerosol source | Slope relative to AE33 REF | | Adjusted $r^2$ | |
|---|---|---|---|---|---|
| | | BM | AM | BM | AM |
| **Group A** | | | | | |
| A01 | Soot | 1.026 | 0.951 | 0.999 | 1 |
| | Nigrosin | 0.934 | 1.025 | 1 | 0.999 |
| | Ambient Air | 0.857 | 0.722 | 0.973 | 0.976 |
| A02 | Soot | 1.006 | 0.994 | 0.995 | 1 |
| | Nigrosin | 1.048 | 1.149 | 1 | 0.999 |
| | Ambient Air | 1.117 | 0.935 | 0.988 | 0.979 |
| **Group B** | | | | | |
| B01 | Soot | 0.905 | 0.970 | 0.999 | 1 |
| | Nigrosin | 0.986 | 1.012 | 1 | 1 |
| | Ambient Air | 0.948 | ND | 0.993 | ND |
| B02 | Soot | 1.047 | 1.043 | 1 | 1 |
| | Nigrosin | 1.027 | 1.049 | 1 | 1 |
| | Ambient Air | 0.965 | 0.902 | 0.992 | 0.94 |
| B03 | Soot | 1.061 | 1.057 | 1 | 0.998 |
| | Nigrosin | 0.988 | 0.968 | 0.999 | 1 |
| | Ambient Air | 0.984 | 1.112 | 0.986 | 0.896 |
| B04 | Soot | 1.003 | 1.024 | 1 | 0.999 |
| | Nigrosin | 0.924 | 0.919 | 0.999 | 1 |
| | Ambient Air | 0.863 | 0.814 | 0.993 | 0.935 |
| B05 | Soot | 1.065 | 1.027 | 1 | 1 |
| | Nigrosin | 1.068 | 1.030 | 0.999 | 1 |
| | Ambient Air | 0.661 | 1.129 | 0.967 | 0.948 |
| B06 | Soot | 1.021 | 0.858 | 1 | 0.996 |
| | Nigrosin | 1.138 | 1.088 | 1 | 0.998 |
| | Ambient Air | 0.814 | 0.750 | 0.987 | 0.933 |
| **Group C** | | | | | |
| C01 | Soot | 0.935 | 0.930 | 1 | 1 |
| | Nigrosin | 0.974 | 1.006 | 1 | 1 |
| | Ambient Air | 0.944 | 0.991 | 1 | 1 |
| C02 | Soot | 0.990 | 0.977 | 1 | 1 |
| | Nigrosin | 0.976 | 1.053 | 1 | 1 |
| | Ambient Air | 1.002 | 1.029 | 1 | 1 |
| C03 | Soot | 1.048 | 1.084 | 1 | 1 |
| | Nigrosin | 1.088 | 1.056 | 1 | 1 |





| Instrument | Aerosol source | Slope relative to AE33 REF | | Adjusted $r^2$ | |
|---|---|---|---|---|---|
| | | BM | AM | BM | AM |
| | Ambient Air | 1.062 | 1.076 | 1 | 1 |
| C04 | Soot | 0.964 | 1.059 | 0.999 | 1 |
| | Nigrosin | 1.063 | 1.106 | 1 | 1 |
| | Ambient Air | 1.059 | 0.993 | 1 | 1 |
| **Group D** | | | | | |
| D01 | Soot | 0.957 | 0.964 | 0.997 | 0.999 |
| | Nigrosin | 1.009 | 0.939 | 0.999 | 1 |
| | Ambient Air | 0.949 | 0.988 | 1 | 0.999 |
| D02 | Soot | 1 | 1.04 | 0.999 | 1 |
| | Nigrosin | 0.986 | 1.031 | 1 | 1 |
| | Ambient Air | ND | 1.030 | ND | 0.999 |
| D03 | Soot | 1.033 | 0.924 | 1 | 1 |
| | Nigrosin | 1.121 | 0.889 | 0.998 | 1 |
| | Ambient Air | 1.172 | 0.940 | 0.999 | 0.999 |
| D04 | Soot | 0.842 | 0.918 | 0.999 | 0.999 |
| | Nigrosin | 0.806 | 0.907 | 0.995 | 1 |
| | Ambient Air | 0.872 | 1.042 | 1 | 0.995 |
| D05 | Soot | 0.892 | 0.977 | 0.999 | 0.999 |
| | Nigrosin | 0.812 | 0.953 | 0.998 | 1 |
| | Ambient Air | 0.957 | 0.984 | 0.999 | 0.999 |
| **Group E** | | | | | |
| E01 | Soot | 0.954 | 0.980 | 0.999 | 1 |
| | Nigrosin | ND | ND | ND | ND |
| | Ambient Air | 1.005 | 1.019 | 0.995 | 0.992 |
| E02 | Soot | 0.883 | 0.998 | 0.998 | 1 |
| | Nigrosin | ND | ND | ND | ND |
| | Ambient Air | 1.15 | 0.930 | 0.996 | 0.993 |
| **Group F** | | | | | |
| F01 | Soot | 1.012 | 1.001 | 1 | 1 |
| | Nigrosin | ND | ND | ND | ND |
| | Ambient Air | 1.069 | 0.865 | 0.996 | 0.994 |
| F02 | Soot | 0.993 | 1.012 | 1 | 1 |
| | Nigrosin | ND | ND | ND | ND |
| | Ambient Air | 1.097 | 0.873 | 0.997 | 0.995 |
| F03 | Soot | 1.031 | 1.044 | 1 | 1 |
| | Nigrosin | ND | ND | ND | ND |
| | Ambient Air | 1.017 | 1.022 | 0.998 | 0.995 |





| Instrument | Aerosol source | Slope relative to AE33 REF | | Adjusted $r^2$ | |
|---|---|---|---|---|---|
| | | BM | AM | BM | AM |
| F04 | Soot | 1.052 | 1.076 | 1 | 1 |
| | Nigrosin | ND | ND | ND | ND |
| | Ambient Air | 1.048 | 1.017 | 0.998 | 0.994 |

BM: Before maintenance. AM: After maintenance. ND: No data.





### 3.3 Instrumental noise

The instrumental noise defined as the single standard deviation of the eBC mass concentration, was calculated with
measurements of dry filtered air (particle-free, RH < 40 %), reported with a time resolution of one-minute. Measurements were performed during $4 \pm 1$ hours in average. The noise dependency on the wavelength was also studied for each one of the twenty-three instruments, as well as the influence of the maintenance activities. The results from this analysis are summarized in Fig. 13 and Table S1 from the supplementary material.

In general, the instrumental noise decreased after service, with more significant changes in the lower and middle
wavelengths. The average noise at 1 min time resolution calculated at 370 nm, ranged from 0.030 µg m$^{-3}$ before maintenance, to 0.023 µg m$^{-3}$ after maintenance, which means a decrease of 32 %. At 660 nm, the average value of the instrumental noise dropped from 0.046 to 0.033 µg m$^{-3}$, implicating a reduction of 40 %, which is the highest average reduction for the 7 wavelengths of the AE33. In the near-IR wavelengths the noise remained almost constant; at 880 nm the average noise did not change, passing from 0.032 to 0.031 µg m$^{-3}$ before and after service, respectively; at 950 nm the
average noise was 0.032 µg m$^{-3}$ before and after maintenance. Larger noise values in the near-UV in multi-wavelength aethalometers have been also found in previous intercomparison exercises (Müller et al., 2011).

From the average concentrations of ambient eBC (880 nm) measured every minute, the calculated instrumental noise represented between 1 and 10 % of the concentrations measured in the urban background in Leipzig.

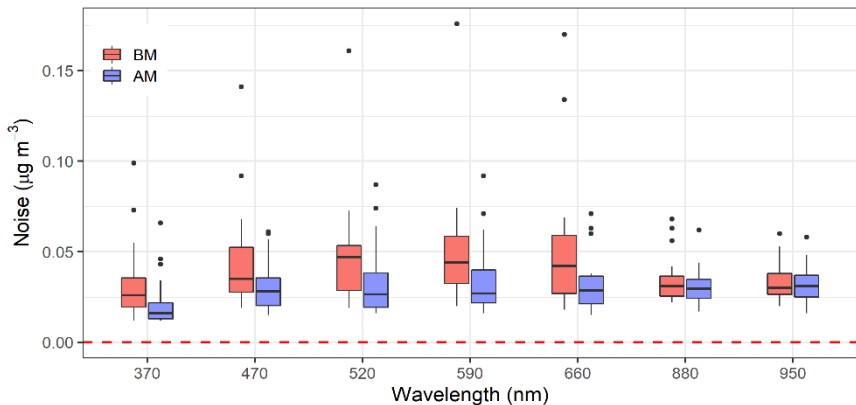


**Figure 13. Average instrument noise at the measurement wavelengths at 1 min time resolution.** The black horizontal line inside the boxes represents the median noise value; the lower and upper borders of the boxes are the first and third quartiles, on which the middle 50% of the noise values are located; the whiskers correspond to ranges for the bottom 25% and the top 25% of the noise without outliers, which are represented by the black points. The red dashed line represents the noise equaling to zero.



### 4. Summary, conclusions and recommendations

In this study, we presented the methodology and results from a comprehensive characterization and intercomparison workshop for aethalometers model AE33 (Magee Scientific). Twenty-three instruments were intercompared at the World Calibration Centre for Aerosol Physics (WCCAP) in Germany, measuring and reporting eBC mass concentrations of laboratory-produced aerosols and ambient air at an urban background site. The instruments received maintenance and were compared against the reference WCCAP aethalometer. The influence of maintenance activities, the filter material and different aerosol sources, in the instrumental variabilities were investigated.

The average unit-to-unit variability in the measurements of eBC mass concentrations (880 nm) reported by the 23 instruments was 1.1 % for soot and 0.3 % for nigrosin, before maintenance. After the maintenance activities, the average variabilities were 0.4 % and -1.1 % for soot and nigrosin, respectively. The average variabilities were calculated using data from measurements performed after one or two filter tape advances to stabilize the internal correction algorithm, as variabilities increased few minutes before a tape advance, when the attenuation is close to 120; in some cases, the offsets among the instruments reached up to 25 %. Tape advances are also crucial since the instruments need to fully adjust to the new aerosol sources and local conditions, and calculate appropriate values of the compensation parameters $k$. The aerosol composition and the filter material exert an influence on the rate of attenuation change and the $k$ values. The combination of these factors directly influences the compensated eBC mass concentrations. Therefore, it is recommended to allow one or two tape advances in the aethalometers before the valid data are obtained for intercomparison purposes, or when the instrument is moved to a new location. For ambient air, the calculation of the total average variability may be biased, as the concentrations measured in the workshops were low (some days < 0.30 µg m$^{-3}$) and stable during the intercomparisons, and unequal among the different groups of instruments. Nevertheless, the results within groups were satisfactory even for very low concentrations measured with a time resolution of one minute.

One of the most important characteristics of the aethalometer AE33 is the reporting of eBC mass concentrations at seven wavelengths. From the intercomparison data analysis, no significant influence of the wavelength in the unit-to-unit variabilities was seen. This fact is important as the spectral range covered by this instrument is usually employed in source apportionment studies. However, the instrumental noise calculated was slightly higher before maintenance in lower (370 nm: 0.030 µg m$^{-3}$, and 470 nm: 0.041 µg m$^{-3}$) and middle wavelengths (660 nm: 0.046 µg m$^{-3}$), and improved significantly after maintenance (370 nm: 0.023 µg m$^{-3}$; 470 nm: 0.031 µg m$^{-3}$; 660 nm: 0.033 µg m$^{-3}$). For higher wavelengths, the instrumental noise was lower and remained almost constant before and after maintenance (880 nm: 0.031 µg m$^{-3}$ and 950 nm: 0.032 µg m$^{-3}$). The instrumental noise was calculated as the single standard deviation of the eBC mass concentrations measured from dry particle-free air; it is important in clean environments with ambient air concentrations similar to those measured during the intercomparisons (0.2 to 3 µg m$^{-3}$). Noise accounted up to 10 % of the average ambient eBC mass concentration reported. The instrumental noise contributes to the uncertainty of the measurements, and must be considered when comparing BC observations intra-sites and inter-sites, in monitoring networks. The unit-to-unit variabilities also contributes to the



uncertainties in the reported eBC inside monitoring networks; these will account for uncertainties associated with instrument flow calibration, leakages, internal correction factors and filter material.

To ensure the measurements of eBC made with filter-based absorption photometers are comparable, reliable and traceable on time, the performance of intercomparisons and maintenance activities are crucial; the type of filter material employed is also very important, as each filter has specific optical properties affecting the measurements of attenuation, used to calculate black carbon mass concentrations. The utilization of a different filter material may result in differences up to 30 %, even after completing the standard maintenance activities. In the AE33 it is strongly recommended to use the most recent version

of filter tape (M8060), and avoid the use of older versions. However, if an aethalometer operates with an older type of filter, it is absolutely necessary to check the use of the corresponding multiple scattering parameter $C$ in the internal settings. This value needs to be checked and confirmed each time the filter tape is changed. Both, the filter type and the correction parameter $C$, should be reported alongside the measurement data when submitting the measurements to databases like EBAS. Besides this, the operators must perform maintenance following the frequencies and instructions given in the user

manual. Flow verification, a leakage test and a check of the spot shape are good starting points to verify the instrument performance. When carrying out a flow verification and calibration, an externally calibrated flowmeter should be used and care should be taken on the flow reporting conditions, otherwise the test and calibration are not reliable. Cleaning the optical chamber and checking the absence of blocking materials in the airflow sample lines, is also vital and should be done more frequently in polluted environments.


*Data availability*. Experiments data will be available at the public data repository EBAS managed by NILU http://ebas.nilu.no/default.aspx, following the data policies from EBAS, COLOSSAL and ACTIRS.

*Supplement*. The supplement related to this article is available online at:


*Author contributions*. LD, ACM, TM, GM, SP and AW planned and designed the study. All co-authors participated in the experiments. ACM processed the data and prepared the manuscript with inputs from LD, TM, GM and AW.

*Acknowledgements*.

→ ACTRIS (European Research Infrastructure for the observation of Aerosol, Clouds and Trace Gases) and COLOSSAL (COST Action CA16109 Chemical On-Line cOmpoSition and Source Apportionment of fine aerosoL), for their financial and logistical support in the preparation and execution of the workshops.

→ The WCCAP for hosting the experiments in their laboratories.

→ Aerosol d.o.o., Slovenia, the manufacturer of the Aethalometer AE33, for troubleshooting support during the

workshops.

→ M. Fernández-Amado acknowledges to Ministerio de Ciencia e Innovación (PTA2017-13607-I).





→    Instrument B04 is operated and maintained thanks to the Spanish Ministry of Science and innovation through the projects CGL2016-81092-R, CGL2017-42 90884REDT and RTI2018.101154.A.I00.

→    G. Titos is funded by Juan de la Cierva-Incorporación postdoctoral program (IJCI-2016-29838).


*Competing interests.* LD and GM have been in the past, but not during the described work or its planning, employed by the manufacturer of the Aethalometer AE33; they are now employed in part by Haze Instruments d.o.o., Slovenia.

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
