# Peer review of "Intercomparison and characterization of 23 Aethalometers under laboratory and ambient air conditions: Procedures and unit-to-unit variabilities"

_Atmospheric Measurement Techniques, 2020_

## Referee Comment (RC1) · Anonymous Referee #1 · 23 Sep 2020

The article presents data from different Aethalometer intercomparisons done in a workshop campaign where lab generated soot, nigrosin and ambient aerosol were used as source of particles for the measurements. A reference Aethalometer is used to assess the unit-to-unit variability. The results of the workshop are analyzed by comparing data before and after applying different maintenance tasks to the compared instruments. The results are valuable for the ambient BC monitoring community and the paper is well written.

I would recommend the publication of the article after addressing the following issues.

[Figure]

**Major comments**

- The authors have mentioned the scattering effect and its compensation constant, C. Not much is detailed in the manuscript about this artifact and they claim this aspect is out of the scope of the study. However, this seems to be a relevant issue for Aethalometers. It would be nice if the authors could provide an estimate of how sensitive the AE is to the different artifacts; filter-loading, scattering by deposited particles, scattering by filter fibers.

- The CAST soot particles seem to have a particle number mean diameter at around 50 nm according to figure 4. This size might be too small for Aethalometer measurements given the particle penetration in the filter. Can the authors comment on that?

- The authors used an AE33 as a reference but it is not clear why this instrument is the reference and how it was calibrated. I encourage the authors to provide details on this.

- When discussing the wavelength-dependency of the unit-to-unit variability the authors should comment on the effects of different artifacts that affect Aethalometer measurements. Since the unit-to-unit variability is only based on a comparison to a reference Aethalometer, the whole wavelength-dependency seem to fit quite well but it is well known that, for example, scattering artifacts will be different at the different wavelengths. The way it is presented might lead to the reader to understand that Aethalometers would be good for retrieving the absorption Angstrom coefficient. Is that the case? Can the authors comment on that? Lines 531-534 is a strong statement that can not be supported with the evidence presented here.

**Minor comments**

l. 87 "When optical methods are used, black carbon is called equivalent black carbon (eBC), because the mass concentration is indirectly retrieved from measurements of light attenuation" Please detail more, why BC measured by FBAP is called equivalent BC?
l. 146 "In the end, it is provided a series of recommendations for operation and maintenance." Please rephrase.

l. 247 Remove the n from "ration".

l. 257 Please avoid starting sentences with an acronym or abbreviation.

l. 278 I encourage the authors to mention the R version they used instead of the IDE version (Rstudio).

l. 305 This filter tape was used when? Before the 2016-2017 tape (M8050)?

l. 392 What about D02 and D05 in the beginning of the comparison? It looks like the deviation is >10%.

Fig. 9 It looks like D01 and D03 performed better before maintenance. Could you please comment on that?

Fig. 9-11 Please use decimal points and not commas in the annotations.

Table 4 The slope values in the table do not seem to be the same ones shown in Fig. 9. Am I missing something?

Please define the acronyms EBAS, EMEP.

---

## Referee Comment (RC2) · Anonymous Referee #2 · 29 Sep 2020

Intercomparisons among AE33 aethalometers were made under well-designed experimental conditions and procedures to quantify the accuracy of their measurement uncertainty, with large efforts from more than 20 research groups. The authors carefully investigated all possible factors affecting on the retrieval of eBC mass concentration, and overall finding and associated discussion are reasonable. This reviewer agrees to its publication in AMT, the following points have to be carefully addressed. This reviewer believes that this paper will greatly help to the people who operate or use data from AE 33.

[Figure]

Abstract: It is highly recommended to highlight key findings of this work, rather than explain the motivation/background of an intercomparison study. Please rewrite the Abstract concisely.

L281: what "diverse situations" means here? I understand that the comparison was performed with various conditions as shown in Figure 5. However, it is necessary to clearly explain the reason(s) why the authors select "Group D" here. This reviewer strongly suggests to present plots (like Figure 5 or Figure 6) for other groups in the supplementary materials.

Figure 7: How is well the spot size area among the aethalometers agreed? If the authors measure, please note.

It is hard to understand whether the aethalometers agree well or not after maintenance from the scatter plots (e.g., Figure 9). In addition to the slope and R2, the reviewer recommends to add BIAS, RMSE and data number in the plot. Other question is how much the aethalometers are disagreed among D01 $\sim$ 05 aethalometers? These questions also go to Figures 10 and 11.

It will be useful if the authors provide same plot of Figure 12 for R2, BIAS, and RMSE. Also, as the authors wrote, it will be helpful if the percent difference is provided. All experimental data are included in Figure 12? Clarify it.

The authors have to clarify that intercomparisons were not simultaneously made for 23 aethalometers.

The authors have to check the typo errors. Also it is needed to check the English by the native speaker.

This reviewer highly expect to see the wavelength-dependency of light absorption from 7-wavelength aethalometer measurements when starting to read the paper. The presentation and discussion of absorption Angstrom exponent (AAE) after calculating absorption coefficient with one of well-known methods will greatly enhance the findings

of this study.

The recommendations given in line 556-559 are simple and all aethalometer users know it, I believe.

---

## Referee Comment (RC3) · Anonymous Referee #3 · 6 Oct 2020

**Cuesta-Mosquera *et al.*: Intercomparison and characterization of 23 Aethalometers under laboratory and ambient air conditions: Procedures and unit-to-unit variabilities.**
**https://doi.org/10.5194/amt-2020-344**

**REVIEW**

The AE33 is used at a large number of measurement sites and applications all over the world. Therefore it is very important that its performance and unit-to-unit variability is tested and presented to the whole community using it. In this paper very many AE33 units were compared which gives credibility to the results. The instruments were compared using both generated soot and nigrosin particles and ambient aerosols. The effects of different maintenance procedures were also analyzed. All of this is important. The experiments were conducted carefully and the paper is well written, I didn't find any obvious errors, I can recommend its publication in AMT. I only have two very minor modification suggestions:

1) L206AMCA? What does this stand for? Why 21.1°C? The most commonly used standard temperatures are 0°C and 25°C.

2) L377-378 " *... From the mathematical definition (Eq. (3) and Eq. (4)) the k values are inversely proportional to eBC, ...*" First, this claim is not intuitively clear from Eqs. (3) and (4). I wish you derived the relationship, for instance this way:

$$b_{abs} = \frac{s(\Delta ATN/100)}{F_1(1-\zeta)C(1-kATN)\Delta t} = \frac{1}{1-kATN}\frac{(\Delta ATN/100)}{F_1(1-\zeta)C\Delta t}$$

Here the last term is the non-compensated absorption coefficient $b_{abs,nc} = \dfrac{s(\Delta ATN/100)}{F_1(1-\zeta)C\Delta t}$

Then the compensation parameter can be calculated as as function of absorption coefficient

$$\Rightarrow b_{abs} = \frac{1}{1-kATN}b_{abs,nc} \Leftrightarrow 1-kATN = \frac{b_{abs,nc}}{b_{abs}} \Leftrightarrow kATN = 1 - \frac{b_{abs,nc}}{b_{abs}}$$

$$k = \frac{1}{ATN}\left(1 - \frac{b_{abs,nc}}{b_{abs}}\right) = \frac{1}{ATN}\left(\frac{b_{abs}-b_{abs,nc}}{b_{abs}}\right)$$

and when the relationship $eBC = b_{abs}/\sigma_{air}$ is used for both $b_{abs}$ and $b_{abs,nc}$:

$$\Rightarrow k = \frac{1}{ATN}\left(\frac{eBC - eBC_{nc}}{eBC}\right),$$ where $eBC_{nc}$ is the non-compensated eBC concentration.

Or if you don't want to write all the steps you could at least write the last equation to support your claim. It shows that for a given ATN, if $eBC > eBC_{nc}$ then k > 0 and k is inversely proportional to eBC. There is no doubt that for the generated BC and nigrosin particles this is the case. However, it should not be written as if this were true for all aerosols. In the ambient aerosol the compensation parameter can also be close to zero or even negative, possibly depending on the coating of particles, as has been noted by (Virkkula et al., 2015; Drinovec et al., 2017; Greilinger et al., 2019).

Virkkula et al.: On the interpretation of the loading correction of the aethalometer, AMT, 8, 4415–4427, 2015

Drinovec et al.: The filter-loading effect by ambient aerosols in filter absorption photometers depends on the coating of the sampled particles, AMT, 10, 1043–1059, 2017.

Greilinger et al.: Evaluation of measurements of light transmission for the determination of black carbon on filters from different station types, Atmos. Environ., 198, 1-11, 2019.

---

## Author Comment (AC1) · 21 Dec 2020

The authors of the manuscript entitled "Intercomparison and characterization of 23 Aethalometers under laboratory and ambient air conditions: Procedures and unit-to-unit variabilities", thank the valuable comments and inputs from the Anonymous Referee #1. All the points expressed were addressed. The answers for each one of your comments are available in the pdf document "Correspondence to referee 1" included in the zip file attached.

Please also note the supplement to this comment:
https://amt.copernicus.org/preprints/amt-2020-344/amt-2020-344-AC1-supplement.zip

---

## Author Comment (AC2) · 21 Dec 2020

The authors of the manuscript entitled "Intercomparison and characterization of 23 Aethalometers under laboratory and ambient air conditions: Procedures and unit-to-unit variabilities", thank the constructive comments given by the Anonymous Referee #2. We have addressed all the concerns you have raised. We have responded to each one of your comments in the pdf document "Correspondence to referee 2" included in the zip file attached.

[Figure]

Please also note the supplement to this comment:
https://amt.copernicus.org/preprints/amt-2020-344/amt-2020-344-AC2-supplement.zip

---

## Author Comment (AC3) · 21 Dec 2020

The authors of the manuscript entitled "Intercomparison and characterization of 23 Aethalometers under laboratory and ambient air conditions: Procedures and unit-to-unit variabilities", acknowledge the valuable feedback given by the Anonymous Referee #3. We have addressed all the concerns raised, the responses to your comments are included in the pdf document "Correspondence to referee 3" in the zip file attached.

Please also note the supplement to this comment:

[Figure]

https://amt.copernicus.org/preprints/amt-2020-344/amt-2020-344-AC3-supplement.zip

---

## Author Response (AR2)

**The present document includes the correspondence to referees #1, #2 and #3, answering each one of their comments and indicating the main changes made in the revised manuscript.**

**Correspondence to Anonymous Referee #1**

The authors of the manuscript entitled "Intercomparison and characterization of 23 Aethalometers under laboratory and ambient air conditions: Procedures and unit-to-unit variabilities", thank the valuable comments and inputs from the Anonymous Referee #1. All the points expressed were addressed. Below, we answer to each one of your comments.

**Major comments**

**Comment 1:** The authors have mentioned the scattering effect and its compensation constant, C. Not much is detailed in the manuscript about this artifact and they claim this aspect is out of the scope of the study. However, this seems to be a relevant issue for Aethalometers. It would be nice if the authors could provide an estimate of how sensitive the AE is to the different artifacts; filter-loading, scattering by deposited particles, scattering by filter fibers.

**Response:** The Aethalometers were "calibrated" using the measurement of transmission of light (the determination of attenuation, ATN) and a measurement of carbon content using Soxhlet extracted filters. The assumption was made that the chemically refractory fraction of the sample (remaining on the filter after Soxhlet extraction) is also the light absorbing fraction – this was the "definition" of black carbon (Gundel et al., 1984). The parameter relating ATN and eBC mass concentration is the mass attenuation cross-section, that is the product of the multiple-scattering parameter $C$ and the mass absorption cross-section (Drinovec et al., 2015).

The value of the multiple-scattering parameter $C$ (as parametrized in Weingartner et al, 2003, and Drinovec et al., 2015) is crucial for the determination of the aerosol absorption coefficient. The separation of the loading effect and the multiple-scattering effect in the filter is arbitrary, to a degree. The quantification of the $C$ value requires a comparison between the Aethalometer and a reference instrument. This has so far proved to be challenging, especially at high single-scattering albedos. Even when the loading effect is corrected for, the comparison slope, often interpreted as the parameter $C$, is influenced by the cross-sensitivity to scattering of all filter

photometers, the particle size effects and their penetration depth into the filter. All of the effects above depend on the filter properties and the sample properties.

This manuscript is focused on the comparability, repeatability and noise of the measurements of the BC mass concentration, the so called eBC. Therefore, we prefer to discuss the measurement in terms of the sensitivity and any potential loss of sensitivity due to the properties of the sample. The potential change of the sensitivity when measuring a sample different from what has been used for the characterization of the instruments is a major source of uncertainty of the Aethalometer measurement. Bernardoni et al (2020) have shown that the wavelength dependence of $C$ is not very large. They have conducted their field campaign in the period of considerable uncertainty in the filter properties and have "played it safe" by using only the published filter with known properties (T60A20, also referred to as M8020). The unpublished results from the manufacturer (as documented in the filter box) seem to indicate that the multiple-scattering parameter for the new filter (M8060) $C=1.39$.

To assess the variation between the instruments, we have used well-defined samples, CAST soot and nigrosin, and then used the instruments to measure ambient air in urban background conditions. This serves well the comparison purpose. For the study of the changes in sensitivity when measuring different samples, focusing especially on BC with different size distributions, we have already carried out some laboratory experiments in the framework of the EMPIR Black Carbon project, using different types of reference instruments which measure the absorption without the interferences on the filter, for example the so-called "extinction minus scattering" (see below, reply to Comment 3), photoacoustic instruments and two different photothermal interferometers (one of which is described in Visser et al., 2020).

We have explicitly addressed the use of different tapes in the manuscript. The change of tape from one (T60A20, also referred to as M8020) type to the current one (M8060) requires the user to change the multiple-scattering parameter $C$ and the leakage factor Zeta. In our experience, this is the most important systematic error in the measurements that the user can make in a field or laboratory measurement campaign. We have included this in the recommendations.

**Comment 2:** The CAST soot particles seem to have a particle number mean diameter at around 50 nm according to figure 4. This size might be too small for Aethalometer measurements given the particle penetration in the filter. Can the authors comment on that?

**Response**: We have used a well-defined sample to characterize and compare the Aethalometers. The size is representative of fresh BC, for example such as would be measured next to a busy street. One of the major considerations was the stable operation of the CAST BC source. We have verified the comparison using ambient measurements. The determination of the sensitivity of Aethalometers to BC of different sizes due to the different penetration into the filter or other artifacts is beyond this work. Please see above (reply to Comment 1) for the range of experiments planned or carried out to assess the influence of the size distribution of the measured BC on the sensitivity of the filter photometers.

**Comment 3:** The authors used an AE33 as a reference but it is not clear why this instrument is the reference and how it was calibrated. I encourage the authors to provide details on this.

**Response:** This is an important remark. To a degree, when comparing instruments of the same design, the choice of the instrument, to which others are compared, is arbitrary. However, the instrument needs to be well characterized and party to a very strict quality control process. The reference AE33 belongs to the WMO-GAW World Calibration Centre for Aerosol Physics (WCCAP); the instrument receives adequate maintenance and is operated with the correct accessories. The flow of this AE33 is calibrated with an externally calibrated flowmeter model 4140 F, TSI Inc. Additionally, the absorption coefficients reported by this reference aethalometer, have been compared with the absorption estimated by a reference set-up from the WCCAP, consisting in one nephelometer Aurora 3000, EcoTech, measuring the aerosols light scattering coefficients, and one CAPS PMex Monitor, Aerodyne Research, Inc, measuring the aerosols optical extinction. The absorption from the reference set up is calculated as *absorption = extinction – scattering*.

We have included complementary information about the performance of the reference aethalometer in the supplementary material (Figure S1), and in the section "2.3 Experimental set-up", which contains now the next paragraph (line 230):

"*The aethalometer AE33 used as reference belongs to the WCCAP; it receives frequent maintenance, and is operated with the correct accessories (filter tape M8060). The flow of this AE33 is calibrated with an externally calibrated flowmeter model 4140 F, TSI Inc. The reported absorption coefficients of this reference AE33, have been compared with the absorption calculated by a reference set up from the WCCAP, consisting in one nephelometer Aurora 3000, EcoTech, measuring the aerosols light scattering coefficients, and one CAPS PMex Monitor, Aerodyne Research, Inc, which measures the aerosols optical extinction. The*

*absorption from the reference set up is calculated as absorption = extinction – scattering, at 450, 525 and 635 nm; the measurements at 450 nm and 635 nm are extrapolated to 470 and 660 nm, respectively. The results from this comparison are shown in Figure S1 in the supplementary material*".

**Comment 4:** When discussing the wavelength-dependency of the unit-to-unit variability the authors should comment on the effects of different artifacts that affect Aethalometer measurements. Since the unit-to-unit variability is only based on a comparison to a reference Aethalometer, the whole wavelength-dependency seem to fit quite well but it is well known that, for example, scattering artifacts will be different at the different wavelengths. The way it is presented might lead to the reader to understand that Aethalometers would be good for retrieving the absorption Angstrom coefficient. Is that the case? Can the authors comment on that? Lines 531-534 is a strong statement that cannot be supported with the evidence presented here.

**Response:** The largest wavelength-dependent uncertainty in the measurements is due to the loading effect – the attenuation (ATN) scales with the mass attenuation cross-section and the loading effect is higher at lower wavelengths. When determining the absorption Ångström exponent from the Aethalometer data, the loading effect needs to be corrected first. Any potential dependence of the multiple-scattering parameter $C$ is taken care of in the second step. The wavelength dependence of $C$ is a smaller artifact than the loading effect (using values from Bernardoni et al., 2020, and scaling them to the new tape M8060 (Yus-Díez et al., in preparation); or carrying the experiments with proper reference instruments, see answer to Comment 1). However, the assumption of a universal $C$ used to determine the wavelength dependence of the absorption coefficient is only an assumption, the dependence needs to be determined with reference measurements (Drinovec et al., in preparation) and in different environments and different single-scattering albedo values (Yus-Díez et al., in preparation). There might not be a universal effective $C$ values, but the cross-sensitivity to scattering (Arnot et al., 2005) and its wavelength dependence (see above) will need to be determined depending on the sample properties using additional measurements.

To quantify further the wavelength dependent unit-to-unit variability, we introduced a new subsection named "3.4 Wavelength-dependency of the light absorption"; it describes the results from the estimation of the AAE using power law fitting for the three aerosol sources measured during the intercomparison.

The section 3.4 includes the following:

**3.4 Wavelength-dependency of the light absorption**

*The absorption Ångström exponents (α) were calculated for soot and ambient air measurements, by applying a power law fitting describing the wavelength (λ) dependency of the aerosol light absorption ($b_{abs}$):*

$$b_{abs} = A\,\lambda^{-\alpha}, \tag{9}$$

*The absorption coefficients $b_{abs}$ were first determined from Eq. 5, using the 5 minute-averaged eBC mass concentrations, and the default values of the mass absorption cross sections ($\sigma_{air}$) used by the AE33 for each wavelength (370 nm: 18.47 $m^2 \cdot g^{-1}$, 470 nm: 14.54 $m^2 \cdot g^{-1}$, 520 nm: 13.14 $m^2 \cdot g^{-1}$, 590 nm: 11.58 $m^2 \cdot g^{-1}$, 660 nm: 10.35 $m^2 \cdot g^{-1}$, 880 nm: 7.77 $m^2 \cdot g^{-1}$, 950 nm: 7.19 $m^2 \cdot g^{-1}$; Magee Scientific, 2018).*

*Figure 14 shows the histograms of α estimated for each instrument in group D and the reference aethalometer, before and after maintenance. During soot measurements (Fig. 14a), the median absorption Ångström exponents before maintenance, ranged from 1.19 to 1.30; after maintenance, the median values fluctuated from 1.21 to 1.29. For ambient air (Fig. 14b), the median α before maintenance varied from 1.43 to 1.77; after maintenance, the median α ranged from 1.34 to 1.4. For both aerosol sources, the variability of the absorption Ångström exponents were reduced after maintenance (soot: Interquartile range IQR before maintenance = 0.08, IQR after maintenance = 0.05; ambient air: IQR before maintenance = 0.10, IQR after maintenance = 0.07). Values of α larger than 1, may indicate the presence of organic compounds in the aerosol particles of soot and ambient air. It has been demonstrated that α is also dependent in the aerosol size and coating (Liu et al., 2018; Virkkula, 2020).*

*The values of the absorption Ångström exponents shown in Fig. 14 were calculated using the absorption from channels 1 to 7 (370 nm to 950 nm); some studies suggest the omission of $b_{abs, 370\ nm}$ reduces the uncertainty in the estimation of the absorption Ångström exponent and their use in source apportion models (Zotter et al., 2017). We have revised the impact of calculating α with measurements from six channels (470 nm to 950 nm), but no significant advantage or improvement was found from the omission of $b_{abs, 370\ nm}$ while calculating α from the samples measured in this study.*

**Minor comments**

**Comment 5:** l. 87 "When optical methods are used, black carbon is called equivalent black carbon (eBC), because the mass concentration is indirectly retrieved from measurements of light attenuation" Please detail more, why BC measured by FBAP is called equivalent BC?

**Response:** In agreement with this comment we have complemented this description from the introduction (line 82) as follows:

"*When optical methods are used, the mass concentration of black carbon is indirectly retrieved from optical measurements of light attenuation caused by the aerosol particles – the determined quantity is equivalent to the mass concentration and therefore called equivalent black carbon (eBC; see Petzold et al., 2013). This method employs an external conversion factor known as the Mass Absorption Cross Section (MAC), to estimate the eBC mass concentrations*".

**Comment 6:** l. 146 "In the end, it is provided a series of recommendations for operation and maintenance." Please rephrase.

**Response:** The last paragraph of the introduction was modified:

"*In this investigation, the authors present the results from the largest intercomparison of aethalometers model AE33, where 23 instruments were characterized and measured BC mass concentrations from three different aerosol sources. The main goal is to determine the unit-to-unit variabilities and their tendencies throughout the spectral range covered by the AE33. Also, we studied the influence of the maintenance activities and accessories used by the instruments on the reported eBC concentrations. In the end, we provide a series of simple recommendations for the instrument operation and maintenance*".

**Comment 7:** l. 247 Remove the n from "ration".

**Response:** Thanks, this typo was corrected.

**Comment 8:** l. 257 Please avoid starting sentences with an acronym or abbreviation.

**Response:** Thanks for this observation, we have checked the document and fixed this.

**Comment 9:** l. 278 I encourage the authors to mention the R version they used instead of the IDE version (Rstudio).

**Response:** Agree. In the subsection "2.5 Data processing and analysis", line 273, we have modified the sentence as follows:

"*The processes of data cleaning and analysis were performed in the software R version 4.0.0*".

**Comment 10:** l. 305 This filter tape was used when? Before the 2016-2017 tape (M8050)?

**Response:** The filter T60A20 was in effect used before the filter M8050, from 2014 to 2016. We have clarified this in the text of the manuscript, now the paragraph reads as:

"*The aethalometer D03 used the T60A20 filter tape (also known as M8020 or AE33-FT), made from TFE-coated glass fibers; this was the first filter used in the AE33 (Drinovec et al., 2015), and was available from 2014 to 2016*".

**Comment 11:** l. 392 What about D02 and D05 in the beginning of the comparison? It looks like the deviation is >10%.

**Response:** Thank you for pointing this out. We have modified the paragraph to mention these initial deviations and its causes. Now the paragraph reads as follows (line 389):

"*The final comparison was performed during two days. As observed in Figure 8a, the deviations among the eBC mass concentrations reported by the instruments reduced significantly for all aerosol sources (<10 %), in comparison with the initial and intermediate comparisons performed in group D. Significant deviations (>10%) were observed only at the beginning of this final stage, after the first tape advance while the instruments measured BC mass from the soot source; these higher deviations are associated with the initial adjustment required by the compensation algorithm to a new aerosol source and the effect of a filter tape advance, as mentioned earlier*".

**Comment 12:** Fig. 9 It looks like D01 and D03 performed better before maintenance. Could you please comment on that?

**Response:** From Figure 9 the relative slopes of the aethalometer D01 before and after service, indicate deviations of 4% and 6%, respectively. For the aethalometer D03, the deviations before and after service were 3% and 7%, respectively. In both instruments, the deviations (how much the slopes differ from 1), are certainly higher after service, even if they are all relatively low (<10%). As mentioned in the manuscript, the correlations shown in Figures 9, 10 and 11 were calculated using measurements performed after one or two tape advances, to

avoid the effects produced by the filter advance and the initial adjustment of the compensation algorithm. To evaluate the deviations, it is also fundamental to observe the complete time series before and after service (Figures 5, 6 and 8), which give a broader perspective of the performance of the instruments; from this analysis it is clear the offsets and deviations are significantly higher before maintenance, more remarkable in the case of the instrument D03.

We consider clarifying this analysis is relevant, therefore we have extended the comments about the specific variabilities shown in the Figures 9, 10 and 11, see line 424.

**Comment 13:** Fig. 9-11 Please use decimal points and not commas in the annotations.

**Response:** We have fixed this error in the revised version.

**Comment 14:** Table 4 The slope values in the table do not seem to be the same ones shown in Fig. 9. Am I missing something?

**Response:** Thanks for this observation, you are correct. The non-corresponding values in the Table 4 were corrected.

**Comment: 15:** Please define the acronyms EBAS, EMEP.

**Response:** We have asked Stephen Platt from NILU (Norwegian Institute for Air Research) taking care of the EBAS database, about the meaning of the acronym; in his response, Stephen explains the name of the database EBAS is no longer an acronym but only the name of the database.

EMEP → European Monitoring and Evaluation Programme. This was included in the revised version.

References

Arnott, W. P., Hamasha, K., Moosmüller, H., Sheridan, P. J., and Ogren, J. A.: Towards aerosol light-absorption measurements with a 7-wavelength aethalometer: Evaluation with a photoacoustic instrument and 3-wavelength nephelometer, Aerosol Science and Technology, 39, 17–29, https://doi.org/10.1080/027868290901972, 2005.

Bernardoni, V., Ferrero, L., Bolzacchini, E., Forello, A. C., Gregorič, A., Massabò, D., Močnik, G., Prati, P., Rigler, M., Santagostini, L., Soldan, F., Valentini, S., Valli, G., and Vecchi, R.:

Determination of Aethalometer multiple-scattering enhancement parameters and impact on source apportionment during the winter 2017–2018 EMEP/ACTRIS/COLOSSAL campaign in Milan, Atmos. Meas. Tech. Discuss. [preprint], doi:10.5194/amt-2020-233, in review, 2020.

Drinovec, L., Močnik, G., Zotter, P., Prévôt, A. S. H., Ruckstuhl, C., Coz, E., Rupakheti, M., Sciare, J., Müller, T., Wiedensohler, A. and Hansen, A. D. A.: The "dual-spot" Aethalometer: An improved measurement of aerosol black carbon with real-time loading compensation, Atmos. Meas. Tech., 8(5), 1965–1979, doi:10.5194/amt-8-1965-2015, 2015.

Gundel, L. A., Dod, R. L., Rosen, H. and Novakov, T.: The relationship between optical attenuation and black carbon, Sci. Total Environ., 36, 197–202, doi:10.1016/0048-9697(84)90266-3, 1984.

Petzold, A., Ogren, J. A., Fiebig, M., Laj, P., Li, S. M., Baltensperger, U., Holzer-Popp, T., Kinne, S., Pappalardo, G., Sugimoto, N., Wehrli, C., Wiedensohler, A. and Zhang, X. Y.: Recommendations for reporting black carbon measurements, Atmos. Chem. Phys., 13(16), 8365–8379, doi:10.5194/acp-13-8365-2013, 2013.

Visser, B., Röhrbein, J., Steigmeier, P., Drinovec, L., Močnik, G., and Weingartner, E.: A single-beam photothermal interferometer for in-situ measurements of aerosol light absorption, Atmos. Meas. Tech. Discuss. [preprint], doi:10.5194/amt-2020-242, in review, 2020.

Weingartner, E., Saathoff, H., Schnaiter, M., Streit, N., Bitnar, B. and Baltensperger, U.: Absorption of light by soot particles: Determination of the absorption coefficient by means of aethalometers, J. Aerosol Sci., 34(10), 1445–1463, doi:10.1016/S0021-8502(03)00359-8, 2003.

**Correspondence to Anonymous Referee #2**

The authors of the manuscript entitled "Intercomparison and characterization of 23 Aethalometers under laboratory and ambient air conditions: Procedures and unit-to-unit variabilities", thank the constructive comments given by the Anonymous Referee #2. We have addressed all the concerns you have raised. Next, we respond to each one of your comments.

**Comment 1:** Abstract: It is highly recommended to highlight key findings of this work, rather than explain the motivation/background of an intercomparison study. Please rewrite the Abstract concisely.

**Response:** We agree with this observation, consequently we decided to rewrite the abstract. The new abstract will be finalized once all the corrections and additional analysis suggested by the reviewers are completed, it will be included in the revised manuscript.

*"Aerosolized black carbon is monitored worldwide to quantify its impact on air quality and climate. Given its importance, measurements of black carbon mass concentrations must be conducted with instruments operating in a quality checked and assured conditions, to generate data which are reliable and comparable temporally and geographically.*

*In this study, we report the results from the largest characterization and intercomparison of filter-based absorption photometers -aethalometers model AE33-, belonging to several European monitoring networks. Under controlled laboratory conditions, a total of 23 instruments measured mass concentrations of black carbon from three well-characterized aerosol sources: synthetic soot, nigrosin particles and ambient air from the urban background of Leipzig, Germany. The objective was to investigate the individual performance of the instruments and their comparability; we analyzed the response of the instruments to the different aerosol sources, and the impact caused by the use of obsolete filter materials and the application of maintenance activities.*

*Differences in the instrument-to-instrument variabilities from eBC concentrations reported at 880 nm were determined before maintenance activities (average deviation from total least square regression: -2.0 %, range: -16 % to 7 %, for soot measurements; average deviation: 0.4 %, range: -15 % to 17 %, for nigrosin measurements), and after they were carried out (average deviation: -1.0 %, range: -14 % to 8 %, for soot measurements; average deviation: 0.5 %, range: -12 % to 15 %, for nigrosin measurements). The deviations are in most of the*

*cases explained by the type of filter material employed by the instruments, the total particles load on the filter and the flow calibration.*

*The results of this intercomparison activity show that relatively small unit-to-unit variability of AE33-based particle light absorbing measurements is possible with well-maintained instruments. It is crucial to follow the guidelines for maintenance activities and the use of the proper filter tape in the AE33, to assure high quality and comparable BC measurements among international observational networks".*

**Comment 2:** L281: what "diverse situations" means here? I understand that the comparison was performed with various conditions as shown in Figure 5. However, it is necessary to clearly explain the reason(s) why the authors select "Group D" here. This reviewer strongly suggests to present plots (like Figure 5 or Figure 6) for other groups in the supplementary materials.

**Response:** The expression "diverse situations" was used to denote the multiple technical reasons responsible of the wide deviations observed in the group D. Only in this group we had simultaneously three aethalometers (D03, D04 and D05) using the two filter tapes no longer recommended (T60A20 and M8050); additionally, the instrument D03 exhibited significative deviations in the flow verification test (~30%), and presented an irregular spot shape before maintenance. The group D was selected as a clear representation of the extreme cases found in real monitoring networks; the deviations observed here allowed us to explain in detail the causes of the unit-to-unit variabilities generally seen among aethalometers.

However, we also consider the expression "diverse situations" is broad, so we have rewritten the paragraph as follows (line 276):

*"Section 3.1 presents a detailed analysis of the instruments characterized in the group D, as a case of study illustrating the wide range of deviations observed in real monitoring networks. In this group we have intercompared aethalometers using three different filter materials, and one of them presented unacceptable results from the flow verification test before maintenance. A summary and analysis of the results obtained for the total 23 units intercompared is given in section 3.2; additionally, the Figures S2 to S11 in the supplementary material, present the time series of the measurements performed by the instruments in the groups A, B, C, E and F, before and after maintenance.".*

The Figures 5 and 6 were reproduced in the supplementary material for the instruments intercompared in the groups A, B, C, E, and F (see Figures S2 to S11).

**Comment 3:** Figure 7: How is well the spot size area among the aethalometers agreed? If the authors measure, please note.

**Response:** During the workshops we did not measure the spot size of the instruments, but we observed their shape, saturation and definition. As an illustration for the reader, the Figure 7 compares the characteristics of the spots formed in the instrument D03 presenting a higher deviation and using an obsolete filter tape, and the instrument D01, with optimal operating conditions and showing a lower deviation.

With the aim of being more specific, we have modified the caption of the Figure 7 as follows: "***Shapes of sample spots observed during maintenance in group D. (a) instrument D03, (b) instrument D01.*** *These instruments used different filter tapes (D03: T60A20, D01: M8060). The spot size was not measured during the workshops. Irregular or diffuse edges of the filter spot can indicate leakage.*"

**Comment 4:** It is hard to understand whether the aethalometers agree well or not after maintenance from the scatter plots (e.g., Figure 9). In addition to the slope and R2, the reviewer recommends to add BIAS, RMSE and data number in the plot. Other question is how much the aethalometers are disagreed among D01 ~ 05 aethalometers? These questions also go to Figures 10 and 11.

**Response:** We agree with the necessity of including additional statistics to better describe the change in the deviations of the instruments before and after maintenance. Therefore, the Figures 9, 10 and 11 were complemented by including also the bias, RMSE and the number of data points.

It is also of interest to represent the deviations among the instruments (D01 ~ 05) – we have prepared three auxiliary figures presenting the regressions from the intercomparison of each couple of instruments, during measurements of soot, nigrosin and ambient air: These three figures are shown in the supplementary material (S12, S13 and S14).

**Comment 5:** It will be useful if the authors provide same plot of Figure 12 for R2, BIAS, and RMSE. Also, as the authors wrote, it will be helpful if the percent difference is provided. All experimental data are included in Figure 12? Clarify it.

**Response:** Agree. Figure 12 is being updated and it includes four subplots: (a) Slope, (b) Coefficient of determination r², (c) BIAS and (d) RMSE. The difference (percentage) is also specified in the boxplots.

Answering your question: Figure 12 includes all the deviations calculated for the 23 instruments from groups A to F. We specify this information in the subsection "Wavelength-dependency of the unit-to-unit variability", as follows:

"*Figure 12 shows the boxplots representing the range of the deviations calculated for the 23 instruments; the figure includes the average values of the slope, bias, RMSE and the coefficient of determination R2, before and after maintenance*".

The name of Figure 12 was also modified: "*Statistics from the unit-to-unit variabilities of the 23 units intercompared with the reference AE33, as function of wavelength for soot, nigrosin and ambient air: (a) slope, (b) R2, (c) bias, (d) RMSE.*".

**Comment 6:** The authors have to clarify that intercomparisons were not simultaneously made for 23 aethalometers.

**Response:** Thanks for this observation. We have modified the first paragraph of the section Materials and Methods (line 146), to better clarify this point:

"*The intercomparisons of aethalometers were conducted in three laboratory workshops at the World Calibration Centre for Aerosol Physics (WCCAP) in Leipzig, Germany. During the first workshop (14th to 25th January 2019), the characterization of seventeen AE33 part of the COST action CA16109 COLOSSAL and ACTRIS (Table 1) was performed. In this first experiment, the instruments were divided in four separated groups (A, B, C, D), due to space limitations in the laboratory which did not permit to perform a simultaneous intercomparison. The instruments from each group completed 2.5 to 3 days of measurements. In the second workshop (7th to 12th June 2019), two aethalometers AE33, designated as group E, were intercompared. Finally, four aethalometers, comprising the group F, were intercompared during the third workshop (18th to 20th June 2019). Instruments in groups E and F does not form part of COLOSSAL, they belong to German research and regional monitoring organizations. The same WCCAP reference instrumentation setup was used in all three workshops*".

The groups to which the instruments belong are also specified in the first column of Table 1.

**Comment 7:** The authors have to check the typo errors. Also, it is needed to check the English by the native speaker.

**Response:** In agreement with this comment, we have review and corrected some typos along the document. The revised manuscript was checked by a native English speaker.

**Comment 8:** This reviewer highly expects to see the wavelength-dependency of light absorption from 7-wavelength aethalometer measurements when starting to read the paper. The presentation and discussion of absorption Angstrom exponent (AAE) after calculating absorption coefficient with one of well-known methods will greatly enhance the findings of this study.

**Response:** We agree, it is important to present and discuss the absorption Ångström Exponents. Accordingly, we have included a new subsection named "3.4 Wavelength-dependency of the light absorption coefficient"; it describes the results from the estimation of the AAE using power law fitting for the measurements of soot and ambient air. We show the results of the absorption Ångström Exponents estimated from each instrument in group D, before and after service. The results for the rest of the aethalometers will be included in the supplementary material.

The section 3.4 includes the following:

*3.4 Wavelength-dependency of the light absorption*

*The absorption Ångström exponents (α) were calculated for soot and ambient air measurements, by applying a power law fitting describing the wavelength (λ) dependency of the aerosol light absorption ($b_{abs}$):*

$$b_{abs} = A \, \lambda^{-\alpha}, \tag{9}$$

*The absorption coefficients $b_{abs}$ were first determined from Eq. 5, using the 5 minute-averaged eBC mass concentrations, and the default values of the mass absorption cross sections ($\sigma_{air}$) used by the AE33 for each wavelength (370 nm: 18.47 $m^2 \cdot g^{-1}$, 470 nm: 14.54 $m^2 \cdot g^{-1}$, 520 nm: 13.14 $m^2 \cdot g^{-1}$, 590 nm: 11.58 $m^2 \cdot g^{-1}$, 660 nm: 10.35 $m^2 \cdot g^{-1}$, 880 nm: 7.77 $m^2 \cdot g^{-1}$, 950 nm: 7.19 $m^2 \cdot g^{-1}$; Magee Scientific, 2018).*

*Figure 14 shows the histograms of α estimated for each instrument in group D and the reference aethalometer, before and after maintenance. During soot measurements (Fig. 14a), the median absorption Ångström exponents before maintenance, ranged from 1.19 to 1.30;*

*after maintenance, the median values fluctuated from 1.21 to 1.29. For ambient air (Fig. 14b), the median α before maintenance varied from 1.43 to 1.77; after maintenance, the median α ranged from 1.34 to 1.4. For both aerosol sources, the variability of the absorption Ångström exponents were reduced after maintenance (soot: Interquartile range IQR before maintenance = 0.08, IQR after maintenance = 0.05; ambient air: IQR before maintenance = 0.10, IQR after maintenance = 0.07). Values of α larger than 1, may indicate the presence of organic compounds in the aerosol particles of soot and ambient air. It has been demonstrated that α is also dependent in the aerosol size and coating (Liu et al., 2018; Virkkula, 2020).*

*The values of the absorption Ångström exponents shown in Fig. 14 were calculated using the absorption from channels 1 to 7 (370 nm to 950 nm); some studies suggest the omission of $b_{abs, 370 nm}$ reduces the uncertainty in the estimation of the absorption Ångström exponent and their use in source apportion models (Zotter et al., 2017). We have revised the impact of calculating α with measurements from six channels (470 nm to 950 nm), but no significant advantage or improvement was found from the omission of $b_{abs, 370 nm}$ while calculating α from the samples measured in this study.*

**Comment 9:** The recommendations given in line 556-559 are simple and all aethalometer users know it, I believe.

**Response:** We had included these basic recommendations considering what we observed during the workshops: some operators were new users and did not know about the basic instrument maintenance; the recommendations given are based on laboratory experiments but may also be extrapolated to field measurements.

References

Zotter, P., Herich, H., Gysel, M., El-Haddad, I., Zhang, Y., Mocnik, G., Hüglin, C., Baltensperger, U., Szidat, S. and Prévôt, A. S. H.: Evaluation of the absorption Ångström exponents for traffic and wood burning in the Aethalometer-based source apportionment using radiocarbon measurements of ambient aerosol, Atmos. Chem. Phys., 17(6), 4229–4249, doi:10.5194/acp-17-4229-2017, 2017.

**Correspondence to Anonymous Referee #3**

The authors of the manuscript entitled "Intercomparison and characterization of 23 Aethalometers under laboratory and ambient air conditions: Procedures and unit-to-unit variabilities", acknowledge the valuable feedback given by the Anonymous Referee #3. We have addressed all the concerns raised. Next, we respond to each one of your comments.

**Comment 1:** L206 AMCA? What does this stand for? Why 21.1°C? The most commonly used standard temperatures are 0°C and 25°C.

**Response:** AMCA stands for Air Movement and Control Association International (amca.org). This is a North American body generating standards for air movement including ventilation and air conditioning; the values of 21.1°C and 1013 hPa, are the air standard temperature and pressure established by the AMCA. These are the default standard conditions used by the flow sensors in the AE33, to report the measured mass flow (Magee Scientific, 2018). We feel it is best to report raw measurements in addition to the processed ones. Although the flow reporting conditions can be modified in the instruments, we assured all the aethalometers used the AMCA in the laboratory as these are the most regularly used conditions in the AE33.

This information was included as a footnote in Table 2.

*"AMCA (Air Movement and Control Association International) are the default standard conditions used by the flow sensors in the AE33, to report the measured mass flow (Magee Scientific, 2018)".*

**Comment 2:** L377-378 " ... From the mathematical definition (Eq. (3) and Eq. (4)) the k values are inversely proportional to eBC, ..." First, this claim is not intuitively clear from Eqs. (3) and (4). I wish you derived the relationship, for instance this way:

$$b_{abs} = \frac{s(\Delta ATN/100)}{F_1(1-\zeta)C(1-kATN)\Delta t} = \frac{1}{1-kATN}\frac{s(\Delta ATN/100)}{F_1(1-\zeta)C\Delta t}$$

Here, the last term is the non-compensated absorption coefficient $b_{abs,nc} = \frac{s(\Delta ATN/100)}{F_1(1-\zeta)C\Delta t}$

Then the compensation parameter can be calculated as a function of absorption coefficient

$$\Rightarrow b_{abs} = \frac{1}{1-kATN}b_{abs,nc} \Leftrightarrow 1-kATN = \frac{b_{abs,nc}}{b_{abs}} \Leftrightarrow kATN = 1 - \frac{b_{abs,nc}}{b_{abs}}$$

$$k = \frac{1}{ATN}\left(1 - \frac{b_{abs,nc}}{b_{abs}}\right) = \frac{1}{ATN}\left(\frac{b_{abs} - b_{abs,nc}}{b_{abs}}\right)$$

And when the relationship $eBC = b_{abs}/\sigma_{air}$ is used for both $b_{abs}$ and $b_{abs,nc}$:

$$\Rightarrow k = \frac{1}{ATN}\left(\frac{eBC - eBC_{nc}}{eBC}\right), \text{ where the } eBC_{nc} \text{ is the non-compensated eBC concentration.}$$

Or if you don't want to write all the steps you could at least write the last equation to support your claim. It shows that for a given ATN, if eBC > eBC_nc then k > 0 and k is inversely proportional to eBC. There is no doubt that for the generated BC and nigrosin particles this is the case. However, it should not be written as if this were true for all aerosols. In the ambient aerosol the compensation parameter can also be close to zero or even negative, possibly depending on the coating of particles, as has been noted by (Virkkula et al., 2015; Drinovec et al., 2017; Greilinger et al., 2019).

**Response:** We totally agree with your comment, the statement "the $k$ values are inversely proportional to eBC" is not naturally seen from equations 3 and 4. We modified the narrative and the derivation according to the reviewer's suggestion. In summary, the description given in section 2.1 explains the algorithm as follows:

1. The attenuation of both spots is calculated as:

$$ATN(\lambda) = -100 * \ln\left(\frac{I}{I_0}\right) \quad (1)$$

2. The compensation parameter is estimated from the proportionality of the loading from both spots, to the airflows F1 and F2:

$$\frac{F_2}{F_1} = \frac{\ln(1 - k * ATN_2(\lambda))}{\ln(1 - k * ATN_1(\lambda))} \quad (3)$$

3. The attenuation is used to estimate the uncompensated absorption:

$$b_{abs}(\lambda)^{non\ comp.} = \frac{s * (\Delta ATN_1(\lambda)/100)}{F_1 * (1 - \zeta) * C * \Delta t} \quad (2)$$

4. The values of $k$ are used to calculate the compensated absorption:

$$b_{abs}(\lambda)^{comp.} = \frac{b_{abs}(\lambda)^{non\ comp.}}{(1 - k(\lambda) * ATN_1(\lambda))} \quad (4)$$

5. The compensated eBC mass concentration is finally estimated using the BC mass absorption cross section:

$$eBC(\lambda)^{comp.} = \frac{b_{abs}(\lambda)^{comp.}}{\sigma_{air}(\lambda)} \qquad (5)$$

In the subsection 3.1, line 373, we have modified the paragraph as follows:

*The k values also depend on the filter type as the different materials determine the filter loading rate, consequently the moment when the threshold attenuation ($ATN_{TA}$) is attained. In addition, the k values are susceptible to the type of aerosols measured (composition and size) and their mixing state (Drinovec et al., 2017). If the equation 4 is rearranged and expressed in terms of eBC, the k values could be defined as a function of the non-compensated and compensated black carbon mass concentration:*

$$k = \frac{1}{ATN}\left(\frac{eBC^{comp.} - eBC^{non\ comp.}}{eBC^{comp.}}\right) \qquad (8)$$

*According to Eq. 8, for a given attenuation, if the compensated eBC is larger than the uncompensated, k will be positive and inversely proportional to the eBC mass; this was observed for the instruments D04 and D05, with higher and positive deviations from our reference aethalometer (Fig. 6b).*